# Bi-directional Bias Attribution: Debiasing Large Language Models without Modifying Prompts

**Yujie Lin**[1]   **Kunquan Li**[1]   **Yixuan Liao**[2]   **Xiaoxin Chen**[2]   **Jinsong Su**[1,3] *

[1]School of Informatics, Xiamen University, China    [2]Vivo AI Lab, China
[3]Key Laboratory of Digital Protection and Intelligent Processing of Intangible Cultural Heritage of Fujian and Taiwan (Xiamen University), Ministry of Culture and Tourism, China
{linyujie, likunquan}@stu.xmu.edu.cn, jssu@xmu.edu.cn

## Abstract

Large language models (LLMs) have demonstrated impressive capabilities across a wide range of natural language processing tasks. However, their outputs often exhibit social biases, raising fairness concerns. Existing debiasing methods, such as fine-tuning on additional datasets or prompt engineering, face scalability issues or compromise user experience in multi-turn interactions. To address these challenges, we propose a framework for detecting stereotype-inducing words and attributing neuron-level bias in LLMs, without the need for fine-tuning or prompt modification. Our framework first identifies stereotype-inducing adjectives and nouns via comparative analysis across demographic groups. We then attribute biased behavior to specific neurons using two attribution strategies based on integrated gradients. Finally, we mitigate bias by directly intervening on their activations at the projection layer. Experiments on three widely used LLMs demonstrate that our method effectively reduces bias while preserving overall model performance. Code is available at the github link: https://github.com/XMUDeepLIT/Bi-directional-Bias-Attribution.

## 1 Introduction

Large language models (LLMs) (Achiam et al., 2023; Dubey et al., 2024) have achieved remarkable performance across a wide range of natural language processing tasks. However, mounting evidence shows that these models can perpetuate and even amplify societal biases, such as gender, racial, religious, and occupational stereotypes (Nadeem et al., 2020). Such biases become especially problematic when LLMs are utilized in critical applications, including content generation, decision-support systems, and interactive dialogues (Liang et al., 2021; Parrish et al., 2021; Gallegos et al., 2024a). As LLMs grow in scale and generalization capacity, understanding and mitigating their internal sources of biased behavior becomes increasingly critical.

During the period of masked language models (Devlin et al., 2019; Liu et al., 2019), some approaches attempted to mitigate bias by fine-tuning models using existing or synthesized datasets (Liang et al., 2020; Guo et al., 2022). However, with the advent of large language models, such methods have become increasingly impractical due to their substantial demands on time and computational resources. To address these limitations, recent efforts primarily focus on prompt-based debiasing, such as explicitly instructing the model to avoid relying on certain biased attributes in its response (Furniturewala et al., 2024), or analyzing the initial output to identify bias patterns before prompting the model to answer again (Gallegos et al., 2024b; Li et al., 2024). Nonetheless, modifying user prompts may negatively impact user experience, especially in multi-turn interactions where repeated rewriting significantly increases context length and inference cost. These challenges motivate us to develop a debiasing approach that requires neither model fine-tuning nor prompt modification.

In this paper, we propose a framework for stereotype cue detection and bias attribution in LLMs, with the goal of identifying biased neurons and applying interventions in an interpretable man-

---
*Corresponding author.

ner. In this work, we define stereotype cues as adjectives or nouns that obviously induce skewed predictions toward specific demographic groups. For example, when no additional gender information is provided in the context, LLMs may tend to associate a doctor with being male; in this case, "*doctor*" serves as a stereotype cue. Our framework is built on two key stages: (i) *Stereotype Cue Selection via Entropy Minimization.* By constructing sentence templates and computing entropy over the model's predicted distribution across demographic groups, we identify the most bias-inducing cues in a model-specific and attribute-specific way. (ii) *Forward and Backward Bias Attribution via Integrated Gradients.* To trace biased outputs back to specific neurons in the LLM, we design two attribution strategies. The *Forward-IG* strategy is to construct prompts in which the subject contains an unknown demographic group, and let the LLM predict the demographic group. Forward-IG quantifies neuron-level bias contributions when the LLM predicts skewed demographic groups from stereotype-laden prompts. Conversely, the *Backward-IG* strategy is to construct a series of sentence subsets, where each subset contains sentences whose subjects belong to different demographic groups, in order to examine the relationship between the model's outputs and demographic information. Backward-IG identifies neurons that drive differences in generated outputs across demographic groups. Overall, these two attribution strategies provide parallel perspectives: Forward-IG captures neuron contributions when the model infers demographic groups from stereotype cues, while Backward-IG highlights neurons responsible for group-dependent disparities in generated text. Together, they enable a comprehensive identification of bias-related neurons that directly shape the model's outputs. After identifying biased neurons, we intervene by fixing their activation values at the projection layer, the final layer before token prediction. By combining attribution and intervention, our framework offers a comprehensive pipeline for debiasing large language models at the neuron level. This contributes to the broader goal of building more trustworthy LLMs.

To summarize, our main contributions are summarized as three-fold:

- We introduce an entropy-based method to identify stereotype cues that elicit biased model behavior, covering both adjective and noun forms.
- We propose Forward-IG and Backward-IG, two gradient-based attribution strategies for identifying neurons responsible for biased generation, respectively. Then we present an effective intervention that directly modifies the projection layer activations, improving fairness with minimal degradation to model performance. Moreover, we theoretically establish the intrinsic connection between bias reduction and output variation.
- We conduct extensive experiments across four demographic attributes using three widely-used LLMs, providing insights into internal bias mechanisms.

## 2 BACKGROUND

In this section, we first decompose debiasing LLMs into two distinct subproblems, and then provide a brief overview of the attribution method IG and its bias attribution variant $IG^2$.

### 2.1 PROBLEM DEFINITION

**Definition 1** (Demographic-Invariant Generation (DIG))**.** *Let $\mathcal{X}$ be the prompt space, $\mathcal{Y}$ the output space, $\mathcal{D}$ the demographic attribute space (e.g., gender, race), and $\theta$ parameterizes a language model inducing $P_\theta(y \mid x)$ over $\mathcal{Y}$. Let $g:\mathcal{D} \to \mathcal{X}$ be a prompt generator that injects demographic information $d \in \mathcal{D}$ into prompts (e.g., "Her mother was very ..." or "Ethiopian men are ..."). We say the model satisfies demographic-invariant generation if:*

$$P_\theta(y \mid g(d)) \approx P_\theta(y \mid g(d')) \quad \forall d, d' \in \mathcal{D}. \tag{1}$$

That means the model's output distribution should remain approximately unchanged when only the demographic information in the prompt varies.

**Definition 2** (Stereotype-Free Inference (SFI))**.** *Let $x \in \mathcal{X}$ be a prompt containing stereotype cues (e.g., words like "doctor", "nurse", "CEO") which may be related to demographic attributes. Given the demographic label space $\mathcal{D}$, the model's conditional prediction of demographic identities from such prompts should not be biased. We say the model can address stereotype-free inference if:*

$$P_\theta(d \mid x) \approx P_\theta(d' \mid x) \quad \forall d, d' \in \mathcal{D}, \tag{2}$$

where $P_\theta(d \mid x)$ is the probability of the model associating $x$ with the demographic group $d$ (e.g., by predicting "man"/"woman" for "The doctor is likely a ..").

In other words, the model should not systematically favor one demographic over another when interpreting stereotype-prone prompts.

## 2.2 Integrated Gradient and Integrated Gap Gradient

This section details two feature attribution methods: *Integrated Gradient* and *Integrated Gap Gradient*, with the latter specifically designed for bias analysis in language models.

**Integrated Gradients (IG)** Sundararajan et al. (2017) attribute model predictions to input features by integrating gradients along a straight path from a input baseline $x'$ to the input $x$. For a model $F : \mathbb{R}^d \rightarrow \mathbb{R}$, the attribution score for the $i$-th feature is

$$\text{IG}(x_i) = (x_i - x_i') \times \int_{\alpha=0}^{1} \frac{\partial F(x' + \alpha(x - x'))}{\partial x_i} d\alpha. \tag{3}$$

$\text{IG}(x_i)$ represents the contribution of the $i$-th input feature to the model's prediction $F(x)$ relative to a baseline $x'$. Here, the term $x - x'$ captures the magnitude of change in the feature from the baseline, while the integral computes the average gradient of the model's output with respect to $x_i$ along the straight-line path between $x'$ and $x$. This approach ensures that the attribution is sensitive to both the scale of the feature variation and the model's response to incremental changes in the input.

**Integrated Gap Gradients (IG$^2$)** Liu et al. (2024) extend the idea of IG to analyze the internal mechanisms responsible for biased behaviors in language models. While IG attributes the output of a model to its input features, IG$^2$ instead attributes the prediction gap between binary demographic pairs (e.g., female vs. male) to internal neurons, enabling the identification of social bias neurons. Formally, given a pair of demographics $d_1$ and $d_2$, and the $j$-th neuron $h_j^{(l)}$ in the $l$-th FFN layer of a model and $h_j^{(l)}$'s initial activation $\overline{h}_j^{(l)}$, IG$^2$ computes the attribution score as

$$\text{IG}^2(h_j^{(l)}) = \overline{h}_j^{(l)} \int_0^1 \frac{\partial \left| P(d_1 \mid \alpha \overline{h}_j^{(l)}) - P(d_2 \mid \alpha \overline{h}_j^{(l)}) \right|}{\partial h_j^{(l)}} d\alpha, \tag{4}$$

where $P(d_i \mid \alpha \overline{h}_j^{(l)})$ denotes the model's prediction probability for demographic $d_i$ when neuron $h_j^{(l)}$ takes the value $\alpha \overline{h}_j^{(l)}$. This formulation directly attributes the difference in model confidence between demographic groups to individual neuron activations, revealing their contribution to biased behavior. These identified neurons, termed social bias neurons, can then be suppressed to mitigate bias without requiring model retraining. However, although IG$^2$ has demonstrated success on masked language models such as BERT (Devlin et al., 2019), applying this neuron-suppression-based debiasing approach to modern large language models still faces several challenges. First, in the lower layers of deep language models, the contribution of individual neuron activations to the final output tends to be marginal, as their influence is increasingly transformed and potentially suppressed by the model's subsequent non-linear operations. This constraint undermines the effectiveness of interventions aimed at modifying the model's token generation probabilities. Second, as shown in Equation 4, IG$^2$ can only capture bias relationships between specific demographic pairs (e.g., predefined biased pairs such as "*driver–doctor*" or "*waiter–lawyer*"). However, cross-pair bias relationships (e.g., between "*driver*" and "*waiter*") are not considered, which may lead to unreliable attribution of model bias. Additionally, a key unresolved issue is how to systematically identify input words that reliably trigger biased responses, as such triggers are crucial for enabling precise bias attribution and improving the efficacy of debiasing strategies. To handle these challenges, our goal is to design an effective solution that jointly tackles the DIG and SFI problems.

## 3 Methodology

In this section, we describe our debiasing method in detail. As shown in Figure 1, our method mainly consists of the following steps: stereotype cue selection and two attribution strategies.

### 3.1 Stereotype Cue Selection

In this work, we define "stereotype cues" as adjectives or nouns that are likely to trigger biased model outputs. Unlike (Guo et al., 2022) where both demographic attributes and stereotype cue

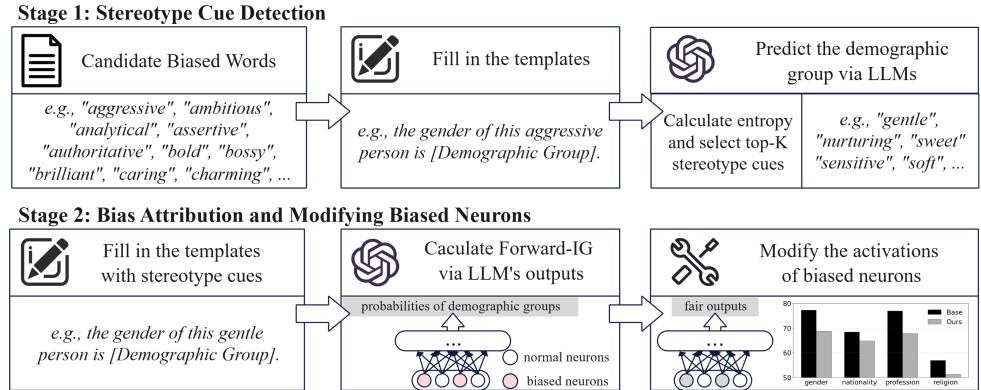

Figure 1: Overview of our method (illustrated with Forward-IG). We first identify the words that trigger biased behavior in the model, then use these words to elicit such behaviors. Based on this, we attribute the biased responses to the most influential neurons and subsequently modify their values. In the bottom-right figure, the gray neurons denote the bias-related neurons after modification. The bar chart presents the debiasing performance of Llama-3.1 on `StereoSet`. The x-axis corresponds to four types of bias, while the y-axis represents the SS score, where values closer to 50% indicate greater fairness. Our method (gray bars) achieves results demonstrates improved fairness.

Table 1: Examples of templates for two types of stereotype cues.

| Category | Template Examples |
|---|---|
| Adjective | The [Demographic_Attribute] of this [Stereotype_Adjective] person is [Demographic_Group]. |
| Noun | The [Demographic_Attribute] of this [Stereotype_Noun] is [Demographic_Group]. |

words are predefined before generating the connecting tokens, our work focuses on automatically identifying the cues that most effectively elicit model biases. Different from (Liu et al., 2024) that defines a set of adjective-based templates, we modify some of these templates and extend them to cover noun-based constructions as well. Table 1 provides examples of the templates, and the complete list can be found in Appendix A.8. We first utilize GPT-4 (Achiam et al., 2023) to help us identify adjectives that are potentially associated with various stereotypes. These adjectives and nouns are then used to construct the candidate list of stereotype cues.

**Entropy-Based Bias Quantification.** The core intuition is that a stereotype cue exhibits stronger bias induction if it causes the model to generate highly skewed predictions in favor of specific demographic groups. We access this via Shannon entropy over the model's conditional probability distribution over demographic groups. Formally, given a candidate stereotype cue $w$ (adjective or noun) and a set of demographic groups $D = \{d_1, d_2, ...\}$, we compute entropy $H(p_{agg})$ where $p_{agg}$ denotes the average of $p(d_i|Replace(t, w))$ computed over all templates. Here, $p(d_i|Replace(t, w))$ represents the model's predicted probability distribution of demographic group $d_i$ given prompts containing $w$ and $Replace(t, w)$ denotes the sentence constructed by inserting the cue $w$ into a predefined template $t$. Lower entropy values indicate more concentrated probability distributions, signaling stronger bias induction by the cue $w$.

**Cue Selection.** The stereotype cue selection process involves four key substages (Appendix A.5), as outlined below: (i) Candidate Pool Initialization. We first collect the candidate lists of adjectives and nouns, ensuring these words are commonly used expressions that are likely to induce model biases. These adjective and noun lists are denoted as $V_{adj}$ (adjectives) and $V_{noun}$ (nouns), respectively. (ii) Probability Collection. For each candidate cue $w \in V_{adj} \cup V_{noun}$, we generate prompts using the templates specified in Table 1 (with [Stereotype_Adjective] or [Stereotype_Noun] placeholders replaced by $w$). For each generated prompt, we query the language model to obtain predicted probabilities over the demographic groups. This is done by constraining generation to the set $D$ and extracting softmax probabilities from the model's final layer. (iii) Aggregate Entropy Calculation. For each cue $w$, we compute the average probability distribution across all templates, and calculate the entropy of this aggregated distribution. This averaging mitigates template-specific noise, ensuring robust bias assessment. (iv) Ranking and Selection. Cues are sorted by their entropy values in the ascending order. We conduct stereotype cue selection on the four demographic attributes in the `StereoSet` dataset. Table 2 reports the top five cues with Llama-3.1 (Dubey et al., 2024).

Table 2: Selected cue examples across four demographic attributes.

| | Demographics | Types of Cues | Top-5 Selected Cues |
|---|---|---|---|
| **Llama-3.1** | Gender | Adjective | gentle, nurturing, sensitive, soft, sweet |
| | | Noun | cheerleader, nurturer, barrier-breaker, confidante, delegate |
| | Nationality | Adjective | warmongering, imperialist, tribal, underdeveloped, primitive |
| | | Noun | commissar, chauvinist, shaman, propagandist, tribesperson |
| | Profession | Adjective | calculating, sensitive, empathic, anxious, emotional |
| | | Noun | robot, assistant, counselor, bureaucrat, sidekick |
| | Religion | Adjective | extremist, ascetic, monastic, evangelical, self-denying |
| | | Noun | meditator, pietist, exorcist, extremist, apostle |

## 3.2 FORWARD BIAS ATTRIBUTION

This paper refers to the causal direction consistent with the SFI problem (from prompts to demographics) as the forward direction, where the input prompt contains stereotype cues and the model is asked to predict which specific demographic group the sample belongs to. The direction associated with the DIG problem is referred to as the backward direction. We first replace the corresponding slots in the templates with the demographic attribute terms and the selected stereotype cues in Section 3.1, generating sentences such as "*The gender of this sensitive person is [Demographic_Group]*". These sentences collectively form a synthetic dataset $DS_f$. For each sample in $DS_f$, we construct a corresponding prompt (see Appendix A.10) and use it to prompt the model to predict the sample's demographic group. To effectively improve the fairness of the model's output probabilities, we attribute the model bias to the input neurons of the projection layer in the LLM (i.e., the layer that maps high-dimensional representations to logits). This allows us to identify bias-related neurons and intervene accordingly. Specifically, for the $j$-th input neuron $h_j$ of the projection layer and its initial activation value $\overline{h}_j$, we propose Forward-IG to quantify the variation in the outputs across all demographic groups for the neuron $h_j$:

$$\text{Forward-IG}(h_j) = \overline{h}_j \int_{\alpha=0}^{1} \frac{\partial \left[ H(p(d_i | \alpha \overline{h}_j)) \right]^{-1}}{\partial h_j} d\alpha, \tag{5}$$

where $H(\cdot)$ denotes the entropy function, and $\alpha \in [0, 1]$ is a scaling variable that gradually changes the value of neuron $h_j$ from 0 to its original activation $\overline{h}_j$. The smaller $H(p(d_i | \alpha \overline{h}_j))$ is, the larger $\left[ H(p(d_i | \alpha \overline{h}_j)) \right]^{-1}$ becomes, indicating that the model is more certain about which specific demographic group the sample belongs to. Such strong certainty toward a particular demographic group reflects the model's bias. By integrating the gradients, Forward-IG accumulates this certainty along the interpolation path, thereby quantifying the contribution of each neuron to biased predictions. Since the integral in Equation (3) cannot be computed analytically, we follow the approach of IG and IG$^2$ and approximate it using a Riemann sum:

$$\text{Forward-IG}(h_j) \approx \overline{h}_j \sum_{k=1}^{n_{step}} \frac{\partial \left[ H(p(d_i | \alpha_k \overline{h}_j)) \right]^{-1}}{\partial h_j} \cdot \frac{1}{n_{step}}, \tag{6}$$

where $\alpha_k = \frac{k}{n_{step}}$ and $n_{step}$ is the number of approximation steps. Note that Forward-IG is computed only for the bias contribution of neurons with respect to a single sample in $DS_f$. Therefore, we identify biased neurons based on the average Forward-IG scores across all samples in $DS_f$. To access this, we first rank all neurons in the descending order according to their average Forward-IG values. Then, we select the top $N = \beta M$ neurons, where $M$ is the total number of neurons in the relevant layer of the model and $\beta \in [0, 1]$ is a proportion parameter. Once the bias neurons are selected, we proceed to disrupt their activation values. In this work, we fix the values of these bias neurons to a constant $C$. Mathematically, for each neuron $h_j$, its updated value $\hat{h}_j$ is defined as

$$\hat{h}_j = \begin{cases} C, & \text{if } h_j \text{ is the top-}N \text{ neurons} \\ \overline{h}_j, & \text{otherwise.} \end{cases}$$

This operation effectively breaks the contribution of these neurons to the model's biased behavior. This approach of disrupting bias neurons' activation values based on the Forward-IG value selection provides a practical way to address both DIG and SFI problems in large language models, which is further validated in the subsequent experimental section.

Table 3: Examples of generated sentence sets for each demographic group.

| Demographics | Demographic Group | Sentence Subset |
|---|---|---|
| Gender | Male, Female | The gender of this [Stereotype_Noun] is male. The gender of this [Stereotype_Noun] is female. |

### 3.3 Backward Bias Attribution

Backward bias attribution identifies biased neurons through induced differences in the model's generation outputs, which occur in response to prompts containing different demographic groups. Specifically, for a given template $t$ and a fixed demographic attribute, we construct a subset of sentences as shown in Table 3, where the placeholder token [Demographic_Group] in $t$ is replaced with $n_d$ demographic groups associated with that attribute. All sentence subsets constitute the dataset $DS_b$, where each subset contains $n_d$ sentences. For each subset in $DS_b$, we construct $n_d$ prompts to predict the stereotypical adjectives or nouns within the sentences. The candidate options are the stereotype cues selected in Section 3.1. We aim to identify the biased neurons responsible for the model producing different probability distributions over stereotypical cues for different groups. For each sentence subset, we compute Backward-IG in the following:

$$\text{Backward-IG}(h_j) = \overline{h}_j \int_{\alpha=0}^{1} \frac{\partial JSD(p_1(w|\alpha\overline{h}_j)), ..., p_{n_d}(w|\alpha\overline{h}_j))}{\partial h_j} d\alpha, \tag{7}$$

where $JSD(\cdot)$ denotes the Jensen-Shannon Divergence and $w$ denotes the stereotype cue. $JSD(\cdot)$ quantifies the divergence among probability distributions over stereotype cues and is formulated in Appendix A.4. The Backward-IG score quantifies how much a neuron's activation contributes to disparities in model outputs across demographic groups, measured via JSD over the predicted distributions of stereotype cues. As in the forward case, we interpolate neuron activations from zero to their original values and accumulate the gradients of JSD along this path to estimate each neuron's contribution. After obtaining Backward-IG scores for all neurons, we calculate average scores over subsets in $DS_b$ and select the top $N$ as biased neurons. Similar to forward attribution, these neurons are then intervened on by fixing their activation values to a constant $C$, effectively neutralizing their influence on group-dependent output variation. Backward-IG also uses Riemann sum approximation.

### 3.4 The Relationship Between Bias Variation and Output Variation: A Theoretical Analysis

**Theorem 1** (Bias Change under Attribution-Guided Modification). *Let $y$ denote the model output of the projection layer $Proj(\cdot)$ and $B : \mathbb{R}^k \to \mathbb{R}$ be a differentiable bias function, such as the reciprocal of entropy or the Jensen–Shannon divergence introduced above. Suppose the hidden representation $h$ is modified along the path:*

$$h(t) = \overline{h} + t\,\Delta h, \quad t \in [0, 1], \tag{8}$$

*with $\Delta h$ defined by attribution-guided projection on a subset of neurons $S$. Then the change in bias satisfies*

$$|\Delta B| \leqslant \|\nabla B(y(0) + \theta\Delta y)\| \cdot \|\Delta y\|, \quad \theta \in [0, 1]. \tag{9}$$

*where $\Delta y = y(1) - y(0)$ and $y(t) = Proj(\overline{h} + t\Delta h) : t \in [0, 1]$ lies along the modification path.*

Equation 9 aligns with our intuition: the larger the $\Delta y$, the greater the upper bound of $\Delta B$. In the extreme case where the model completely loses its modeling capability, it produces random outputs for inputs from any demographic group, thereby exhibiting minimal bias. Specifically, the bias change $\Delta B$ equals the directional projection of the output shift $\Delta y$ onto the local bias gradient $\nabla B(y(0) + \theta\Delta y)$. A detailed proof of Theorem1 can be found in Appendix A.6.

## 4 Experiments

In this section, we conduct experiments on two debiasing strategies. For brevity, we refer to the strategies based on Forward-IG and Backward-IG as forward bias attribution (FBA) and backward bias attribution (BBA), respectively.

Table 4: Similarity between selected cues or all candidates and gender-associated vocabularies.

| | **Llama-3.1** | | | **Llama-3.2** | | |
|---|---|---|---|---|---|---|
| **Adjective** | $\text{Sim}_m$ (%) | $\text{Sim}_f$ (%) | Diff (%) | $\text{Sim}_m$ (%) | $\text{Sim}_f$ (%) | Diff (%) |
| Top-5 Selected Cues | 8.92 | 4.60 | 4.32 | 6.04 | 7.21 | 6.44 |
| All Candidates | 6.37 | 4.68 | 2.79 | 7.76 | 11.07 | 6.11 |
| **Noun** | $\text{Sim}_m$ (%) | $\text{Sim}_f$ (%) | Diff (%) | $\text{Sim}_m$ (%) | $\text{Sim}_f$ (%) | Diff (%) |
| Top-5 Selected Cues | 7.60 | 6.97 | 3.36 | 8.23 | 15.07 | 9.41 |
| All Candidates | 7.37 | 5.60 | 2.89 | 8.55 | 12.14 | 7.20 |

## 4.1 EXPERIMENTAL SETTINGS

We conduct experiments on three widely used large language models, Llama3.1-8B (Llama-3.1), Llama3.2-3B (Llama-3.2) (Dubey et al., 2024) and Mistral-7B-v0.3 (Mistral-v0.3) (Jiang et al., 2023). Biased neurons are identified and perturbed using Forward-IG and Backward-IG, respectively. We evaluate the effectiveness of both attribution methods on the DIG and SFI tasks. Due to space constraints, all results of Mistral-v0.3 are presented in Appendix A.7.

**Baseline Methods.** We categorize the baselines into three types: (i) a training-based method: Auto-Debias (Guo et al., 2022); (ii) prompt engineering-based methods, including Prefix Prompting (Furniturewala et al., 2024), Self-Debiasing (Gallegos et al., 2024b), and DDP (Li et al., 2024); and (iii) a neuron attribution-based method: IG$^2$ (Liu et al., 2024). Detailed descriptions of all baseline methods are provided in Appendix A.3.

## 4.2 EVALUATION OF STEREOTYPE CUE SELECTION

We design the stereotype–cue selection procedure by grounding it in the target model's own embedding space rather than relying on preconceived human assumptions about stereotypical language. Using gender bias as an illustrative case, we define the male-associated vocabulary as $\mathcal{W}_m = \{\text{"male"}, \text{"man"}\}$, and the female-associated vocabulary as $\mathcal{W}_f = \{\text{"female"}, \text{"woman"}\}$. For the top five terms identified by the stereotype–cue selection method, we compute their average cosine similarity with the male and female vocabularies, denoted by $\text{Sim}_m$ and $\text{Sim}_f$, respectively. We additionally calculate the average absolute difference between these two similarities,

$$\text{Diff} = \frac{1}{N_c} \sum_{i=1}^{N_c} \big|\cos(e_i, \mathcal{W}_m) - \cos(e_i, \mathcal{W}_f)\big|,$$

where $e_i$ denotes the embedding of the $i$-th selected term and $N_c$ debotes the number of candidate words. It can be observed in Table 4 that the selected terms exhibit a larger Diff, indicating that each term is more strongly associated with either male- or female-related vocabulary. This suggests that our cue–selection method effectively identifies terms that elicit more biased model behavior.

## 4.3 DIG TASK (STEREOSET)

StereoSet (Nadeem et al., 2020) is a benchmark designed to evaluate stereotypical bias in pretrained language models. It covers four demographic domains: gender, nationality, profession, and religion. The dataset consists of original sentence and option triplets, each composed of (i) a stereotype option that reflects a common social bias, (ii) an anti-stereotype option that counters the bias, and (iii) an unrelated option that is semantically incoherent or irrelevant. The primary task is to assess whether a language model favors stereotypical content over unbiased or unrelated alternatives, thereby quantifying bias while controlling for contextual relevance. For example, a tuple from the gender domain might include:

Original template: *Girls tend to be more [BLANK] than boys.*
**Stereotype:** *soft*   **Anti-Stereotype:** *determined*   **Unrelated:** *fish*

**Evaluation Metrics.** We employ three complementary metrics: SS, LMS, and ICAT. SS measures the proportion of instances where the model prefers the stereotype option over the anti-stereotype

Table 5: Evaluation results across four demographic attributes on the `StereoSet` dataset.

| Llama-3.1 | Gender | Nationality | Profession | Religion |
|---|---|---|---|---|
| | SS (%) → 50% / LMS (%) ↑ / ICAT (%) ↑ | | | |
| Base | 77.34 / 100.0 / 45.31 | 68.43 / 98.13 / 61.95 | 76.96 / 97.53 / 44.94 | 56.94 / 91.14 / 78.48 |
| Auto-Debias | 79.69 / 100.0 / 40.62 | 69.07 / 98.13 / 60.71 | 77.33 / 98.02 / 44.44 | 70.08 / 91.14 / 54.81 |
| Prefix Prompting | 81.89 / 99.22 / 35.94 | 65.67 / 97.51 / 66.94 | 73.42 / 97.73 / 51.85 | 54.79 / 92.41 / 83.54 |
| Self-Debiasing | 77.97 / 92.19 / 40.63 | 65.24 / 92.10 / 64.03 | 69.17 / 92.10 / 56.79 | 63.38 / 89.87 / 65.82 |
| DDP | 77.17 / 99.22 / 45.31 | 65.45 / 96.88 / 66.94 | 73.91 / 96.54 / 50.37 | 64.18 / 84.81 / 60.76 |
| IG$^2$ | 77.34 / 100.0 / 45.31 | 69.64 / 97.92 / 59.46 | 76.52 / 97.78 / 45.93 | 61.64 / 92.41 / 70.89 |
| FBA | 68.75 / 100.0 / **62.50** | 64.88 / 97.09 / 68.19 | 67.87 / 96.05 / **61.73** | 51.35 / 93.67 / **91.14** |
| BBA | 69.84 / 98.44 / 59.38 | 63.18 / 95.43 / **70.27** | 71.58 / 95.56 / 54.32 | 49.31 / 92.41 / **91.14** |

| Llama-3.2 | Gender | Nationality | Profession | Religion |
|---|---|---|---|---|
| | SS (%) → 50% / LMS (%) ↑ / ICAT (%) ↑ | | | |
| Base | 82.40 / 97.66 / 34.38 | 66.31 / 96.88 / 65.28 | 70.89 / 97.53 / 56.79 | 57.89 / 96.20 / 81.01 |
| Auto-Debias | 80.00 / 97.66 / 39.06 | 65.59 / 96.67 / 66.53 | 70.38 / 97.53 / 57.78 | 59.21 / 96.20 / 78.48 |
| Prefix Prompting | 79.20 / 97.66 / 40.62 | 65.08 / 95.84 / 66.94 | 67.69 / 96.30 / 62.22 | 56.58 / 96.20 / 83.54 |
| Self-Debiasing | 73.11 / 92.97 / 50.00 | 72.17 / 88.15 / 49.06 | 63.94 / 87.65 / 63.21 | 64.06 / 81.01 / 58.23 |
| DDP | 77.17 / 99.21 / 45.31 | 70.57 / 97.51 / 57.38 | 71.13 / 97.53 / 56.30 | 63.51 / 93.67 / 68.35 |
| IG$^2$ | 44.71 / 66.41 / 59.38 | 64.29 / 96.05 / 68.61 | 68.45 / 97.04 / 61.23 | 55.26 / 96.20 / 86.08 |
| FBA | 69.60 / 97.66 / 59.38 | 58.52 / 95.22 / **79.00** | 61.88 / 94.57 / 72.10 | 51.95 / 97.46 / **93.67** |
| BBA | 67.46 / 98.44 / **64.06** | 62.93 / 96.47 / 71.52 | 62.21 / 96.05 / **72.59** | 53.33 / 94.94 / 88.61 |

one. A higher SS indicates a stronger tendency to favor stereotypical associations. Ideally, a fair model should have an SS close to 50%, suggesting no systematic preference for either stereotypes or anti-stereotypes. LMS evaluates the model's ability to prefer meaningful content over incoherent or irrelevant options. A higher LMS reflects better language modeling capability. A desirable model should have an LMS close to 100%, indicating that it consistently favors contextually relevant completions over unrelated ones. ICAT integrates both fairness and fluency by rewarding models that maintain low stereotype bias while preserving high linguistic coherence. The optimal ICAT score is 100%, which would indicate perfect fairness and fluency.

**Overall Performance.** From Table 5, we observe that on Llama-3.1, baseline methods are largely ineffective in mitigating model bias and even exacerbate it in certain domains. On Llama-3.2, some prompt modification approaches begin to show partial effectiveness, but only Prefix Prompting consistently reduces bias across all domains, and the reduction is marginal. Notably, the IG$^2$ method is effective only on Llama-3.2, where it substantially lowers LMS in the gender domain to bring SS closer to 50%. However, such behavior is unacceptable in practical applications. In contrast, FBA and BBA achieve effective bias reduction while incurring little to no loss in modeling capability, ultimately yielding the best overall performance as reflected by the highest ICAT scores.

## 4.4 DIG TASK (`BBQ`)

`BBQ` (Parrish et al., 2021) is a large-scale benchmarking resource designed to evaluate social bias and robust reasoning in question-answering (QA) systems. Developed to probe how QA models handle sensitive demographic attributes, BBQ focuses on whether models rely on stereotypical assumptions or demonstrate contextually grounded reasoning.

**Evaluation Metrics.** BBQ includes two types of questions: those posed under *ambiguous* context and those under *disambiguated* context. In the ambiguous setting, models are expected to select the "unknown" option rather than rely on demographic stereotypes. In the disambiguated setting, models should choose the correct answer based on the explicit contextual evidence. Evaluation is conducted using accuracy on the ambiguous questions ($Acc_{amb}$) and on the disambiguated questions ($Acc_{dis}$). Since `BBQ` does not include a dedicated profession domain, we conduct evaluations on the gender, nationality, and religion domains. The results for each domain are provided in the appendix, and here we report the averaged performance across these three domains. BBQ includes two types of questions: those posed under *ambiguous* context and those under *disambiguated* context. In the ambiguous setting, models are expected to select the "*unknown*" option rather than rely on demographic stereotypes. In the disambiguated setting, models should choose the correct answer based

Table 6: Evaluation results on the BBQ dataset.

| | Llama-3.1 | | | Llama-3.2 | | |
|---|---|---|---|---|---|---|
| | $\text{Acc}_{amb} \uparrow$ | $\text{Acc}_{dis} \uparrow$ | Average $\uparrow$ | $\text{Acc}_{amb} \uparrow$ | $\text{Acc}_{dis} \uparrow$ | Average $\uparrow$ |
| Base | 54.50 | 91.18 | 72.84 | 58.95 | 84.61 | 71.78 |
| Auto-Debias | 54.61 | 91.06 | 72.84 | 58.53 | 84.62 | 71.58 |
| Prefix Prompting | 75.13 | 86.08 | 80.61 | 82.81 | 69.03 | **75.92** |
| Self-Debiasing | 87.73 | 73.02 | 80.38 | 66.06 | 74.17 | 70.11 |
| DDP | 56.87 | 89.93 | 73.40 | 60.38 | 83.40 | 71.89 |
| $IG^2$ | 59.00 | 89.46 | 74.23 | 52.61 | 75.84 | 64.22 |
| FBA | 66.45 | 88.87 | 77.66 | 68.78 | 81.59 | 75.19 |
| BBA | 74.46 | 88.95 | **81.70** | 68.08 | 79.95 | 74.02 |

Table 7: Evaluation results on the WinoBias dataset.

| | Llama-3.1 | | | | Llama-3.2 | | | |
|---|---|---|---|---|---|---|---|---|
| | $P_{stereo}$ | $P_{anti}$ | $P_{other} \downarrow$ | $Gap \downarrow$ | $P_{stereo}$ | $P_{anti}$ | $P_{other} \downarrow$ | $Gap \downarrow$ |
| Base | 62.63 | 37.37 | 0.00 | 25.26 | 95.71 | 4.29 | 0.00 | 91.42 |
| Auto-Debias | 61.49 | 38.51 | 0.00 | 22.98 | 96.46 | 3.54 | 0.00 | 92.92 |
| Prefix Prompting | 58.08 | 41.92 | 0.00 | 16.16 | 96.21 | 3.79 | 0.00 | 92.42 |
| Self-Debiasing | 85.61 | 14.39 | 0.00 | 71.22 | 88.64 | 11.11 | 0.25 | 77.53 |
| $IG^2$ | 57.32 | 42.68 | 0.00 | 14.64 | 43.18 | 43.43 | 13.39 | **0.25** |
| FBA | 52.53 | 47.47 | 0.00 | 5.06 | 59.34 | 40.66 | 0.00 | 18.68 |
| BBA | 50.51 | 49.49 | 0.00 | **1.02** | 51.52 | 48.48 | 0.00 | 3.04 |

on the explicit contextual evidence. Evaluation is conducted using accuracy on the ambiguous questions ($\text{Acc}_{amb}$) and on the disambiguated questions ($\text{Acc}_{dis}$). Since BBQ does not include a dedicated profession domain, we conduct evaluations on the gender, nationality, and religion domains. The results for each domain are provided in Appendix A.7, and here we report the averaged performance across these three domains.

**Overall Performance.** As shown in Table 6, prompt-modification approaches (e.g., Prefix Prompting, Self-Debiasing) boost $\text{Acc}_{amb}$ but significantly undermine model accuracy when the context is unambiguous, resulting in a sharp drop in $\text{Acc}_{dis}$. In comparison, FBA and BBA reduce bias effectively in ambiguous scenarios with only minimal impact on $\text{Acc}_{dis}$. This indicates that our method is more moderate: when clear contextual guidance is available, it still tends to produce the correct answer rather than forcibly selecting the debiased "*unknown*" option.

### 4.5 SFI TASK (WINOBIAS)

WinoBias (Zhao et al., 2018) is a dataset designed to measure gender bias. We adopt a cloze-style version of WinoBias to evaluate the SFI task[1]. Specifically, WinoBias requires the model to predict the demographic subject (e.g., he or she) or modifier in sentences like "*The developer argued with the designer because [MASK] did not like the design*". Specially, since DDP's prompt construction is not applicable to this dataset, it is not adopted in the SFI task.

**Evaluation Metrics.** Similar to StereoSet, WinoBias also provides a stereotype option and an anti-stereotype option. We denote the probabilities of the model selecting these two options as $P_{stereo}$ and $P_{anti}$, respectively, while $P_{other} = 1 - P_{stereo} - P_{anti}$ represents the probability of selecting any other token. A lower $P_{other}$ indicates better language modeling capability, and a smaller gap between $P_{stereo}$ and $P_{anti}$ reflects greater fairness in the model's behavior.

**Overall Performance.** Table 7 shows that BBA, while keeping $P_{other} = 0$ (i.e., without sacrificing language modeling capability), achieves the best and second-best gap values on the two models, respectively. We find that, compared with the DIG task, $IG^2$ demonstrates promising effectiveness on the SFI task. However, a limitation is that, on Llama-3.2, although $IG^2$ reduces the gap nearly to 0, it results in a substantially higher $P_{other}$ value, indicating a severe degradation in the model's language

---

[1]https://huggingface.co/datasets/sasha/wino_bias_cloze1

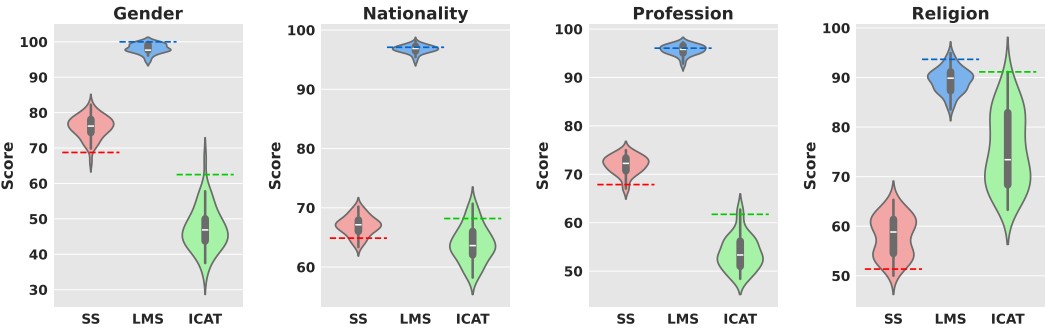

(a) Results when neurons are randomly selected and modified according to the FBA modification ratio. The dashed line denotes the FBA's results.

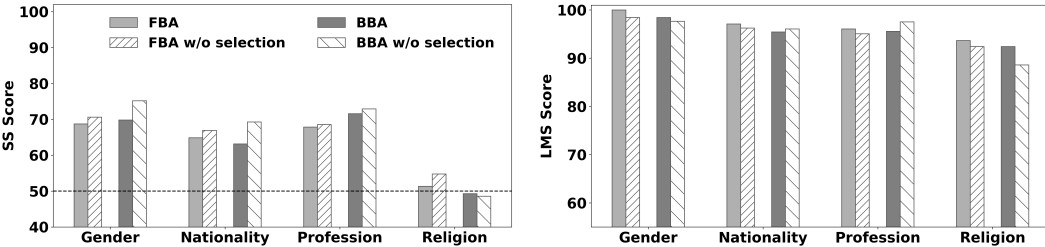

(b) FBA results without stereotype cue detection. The dashed line in the left panel indicates the ideal value.

Figure 2: Ablation results of Llama-3.1 on `StereoSet`: (a) w/o attribution, and (b) w/o selection.

modeling capability. We further find that BBA appears more suitable than FBA for addressing the SFI problem, and it does not exhibit a clear advantage in the DIG task.

## 4.6 ABLATION STUDY

We conducted two types of ablation studies for FBA and BBA: (i) **w/o attribution**. Removing the attribution strategy, where neurons in the projection layer are randomly selected for intervention under the same hyperparameter settings as our method; and (ii) **w/o selection**. Removing the stereotype cue selection algorithm, where candidate words are grouped and the first word of each group is chosen as stereotype cues. In the main text, we only report the results of the FBA method on Llama-3.1. The results for the other two models as well as for the BBA method are provided in Appendix A.7.

**W/o attribution.** To mitigate the unreliability caused by randomly selecting neurons, we conducted 50 tests for the samples in each domain, as shown in Figure 2a. The violin plots illustrate the results of random neuron selection, while the dashed line on each violin corresponds to the FBA results. Except for the profession domain, FBA achieves a win-win outcome compared to random selection, namely SS values closer to 50% and higher LMS. In the profession domain, FBA exhibits only a marginal decline in LMS, while still obtaining substantial debiasing gains.

**W/o selection.** As shown in Figure 2b, without selecting the stereotype cues, SS values overall deviate further from 50%, indicating that our selection algorithm successfully identifies words that are more likely to trigger biased model outputs. Surprisingly, in the absence of the selection algorithm, LMS also shows a slight decrease. This demonstrates that the stereotype cue selection algorithm does not negatively impact the model's language modeling capability.

## 5 CONCLUSION

We presented a neuron-level debiasing framework for large language models that integrates stereotype cue detection, gradient-based bias attribution, and targeted projection-layer intervention. Our approach mitigates demographic bias without requiring fine-tuning or any modification to user prompts, while preserving core language modeling capabilities. By showing that biased behaviors can be localized to specific, identifiable subsets of neurons, our work offers a practical and interpretable pathway toward building fairer and more trustworthy LLMs.

## ACKNOWLEDGEMENT

The project was supported by Natural Science Foundation of Fujian Province of China (No.2024J011001) and the Public Technology Service Platform Project of Xiamen (No.3502Z20231043). We also thank the reviewers for their insightful comments.

## ETHICS STATEMENT

This work examines social bias in large language models. The analysis may involve examples containing stereotypes or sensitive content, which are used solely for research purposes. Our aim is to understand and mitigate bias, not to reinforce it.

## REPRODUCIBILITY STATEMENT

We provide an anonymous link to our code in the abstract, and the proof of Theorem 1 is included in Appendix A.6.

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

CONTENTS

# A    APPENDIX

## A.1    USE OF LLMS.

For each demographic attribute, we employed GPT-4 (Achiam et al., 2023) to assist in generating potentially biased words. In addition, we used LLMs to check the manuscript for typographical errors.

## A.2    RELATED WORKS

**Social Bias in LLMs.** Unlike the bias between the training and test distributions (Wang et al., 2023; 2025), this paper primarily investigates social bias (Zhao et al., 2021; 2022; Lin et al., 2023; Shao et al., 2024; Lin et al., 2025; 2024). Early studies revealed biased associations in distributional representations (e.g., gendered analogies in word embeddings), and proposed geometric debiasing methods to reduce such effects (Bolukbasi et al., 2016). Subsequent task-specific benchmarks exposed bias in core NLP systems. For example, gendered errors in coreference resolution and occupation classification highlighted how downstream models can reproduce and even magnify societal imbalances (Zhao et al., 2018; De-Arteaga et al., 2019). Work focused on generation showed that open-ended models produce differing "regard" and disparate toxicity across demographic groups, and large-scale prompt-based evaluations revealed neural models' propensity for toxic degeneration under realistic prompts (Sheng et al., 2019; Gehman et al., 2020). To quantify stereotyping more broadly, community benchmarks such as CrowS-Pairs and StereoSet were introduced; evaluations on these datasets demonstrate that both masked and autoregressive LMs often prefer stereotyped continuations (Nangia et al., 2020; Nadeem et al., 2020).

Beyond empirical measurement, critical analyses highlight that the scale, opacity, and data practices of modern LLMs generate significant socio-technical risks. These range from environmental and labor concerns to the reproduction of harmful narratives, thereby motivating calls for greater transparency, staged release strategies, and more comprehensive evaluation protocols (Bender et al., 2021). More recent work has expanded the scope of bias evaluations: large-scale audits show that generative models systematically encode social identity biases across dozens of systems (Hu et al., 2025), and new resources such as WinoIdentity allow for fine-grained assessment of intersectional stereotypes across multiple demographic attributes (Khan et al., 2025). Complementary test suites (e.g., BEATS) propose unified frameworks to assess bias and fairness in conjunction with factuality and safety, reflecting the need for multidimensional evaluations of LLM behavior (Abhishek et al., 2025). At the same time, mitigation research has moved beyond prompt editing and fine-tuning. For instance, socially grounded approaches inspired by the contact hypothesis simulate intergroup exposure to reduce biased outputs (Raj et al., 2024), while multi-agent causal intervention frameworks seek to minimize stereotyping without degrading task performance (Xu et al., 2025). Together, these lines of inquiry highlight both the persistence of social bias in modern LLMs and the growing sophistication of evaluation and mitigation strategies.

**Fairness-aware Learning.** Parallel to work documenting bias in LLMs, the broader machine learning community has developed a rich body of research on fairness-aware learning. Early studies formalized group fairness criteria such as demographic parity, equalized odds, and calibration, and explored algorithmic strategies to balance predictive performance with fairness constraints (Dwork

et al., 2012; Hardt et al., 2016; Pleiss et al., 2017). Subsequent research introduced individual fairness notions grounded in similarity metrics, emphasizing that similar individuals should receive similar outcomes (Dwork et al., 2012; Joseph et al., 2016). Beyond static definitions, scholars highlighted tensions among fairness criteria, impossibility results, and trade-offs with accuracy, motivating the development of context-sensitive approaches (Kleinberg et al., 2016; Barocas et al., 2023). To address distributional challenges, fairness-aware methods increasingly account for domain shifts and long-tail groups. In particular, techniques such as reweighting, adversarial learning, and causal inference have been proposed to achieve robust fairness under covariate shift and label imbalance (Saunders & Byrne, 2020; Wu et al., 2019; Garg et al., 2019). More recent directions extend fairness considerations to large-scale generative systems: approaches include counterfactual data augmentation, representation regularization, and fairness-constrained decoding strategies tailored for pre-trained LMs (Saunders & Byrne, 2020; Sheng et al., 2021; Solaiman & Dennison, 2021). At the same time, interdisciplinary critiques stress that fairness cannot be reduced to quantitative metrics alone; fairness-aware learning must also grapple with the structural and sociocultural dimensions of algorithmic decision-making (Selbst et al., 2019; Blodgett et al., 2020).

## A.3 SIMPLE INTRODUCTION FOR OUR BASELINES

**Auto-Debias** (Guo et al., 2022) is a two-stage fine-tuning method for masked language models (MLMs). Without external corpora, the approach uses beam search to automatically discover prompts that maximally expose gender or racial bias in cloze-style completions.

**Prefix Prompting** (Furniturewala et al., 2024) uses simple instructions or role-play prefixes that ask the model to be fair.

**Self-Debiasing** (Gallegos et al., 2024b) ask the model to explain which answer choices rely on invalid assumptions before answering

**DDP** (Li et al., 2024) develops a causality-guided prompting framework. A causal graph models how selection mechanisms in training data create spurious dependencies between social category and model decisions.

## A.4 JENSEN-SHANNON DIVERGENCE

For a set of probability distributions $\{p_1, p_2, \ldots, p_n\}$ over the same probability space, Jensen-Shannon Divergence (JSD) is defined as

$$JSD(p_1, \ldots, p_n) = \frac{1}{n} \sum_{i=1}^{n} KL\Big(p_i \ \Big\| \ \frac{p_1, p_2, \ldots, p_n}{n}\Big), \tag{10}$$

where $KLD(\cdot)$ denotes the Kullback-Leibler divergence.

## A.5 STEREOTYPE CUE SELECTION ALGORITHM

**Cue Selection Pipeline.** The process consists of four steps:

- Candidate Initialization: Collect adjective and noun sets $V_{adj}, V_{noun}$.
- Probability Collection: Generate prompts with templates and obtain $p(d_i|\cdot)$ from the model.
- Entropy Calculation: Aggregate probabilities across templates and compute entropy.
- Ranking and Selection: Rank cues by entropy (ascending) and select those with strongest bias induction.

## A.6 DERIVATION OF THEOREM 1

### A.6.1 DEFINITIONS AND PRELIMINARIES

**Assumptions.** We assume that $Proj : \mathbb{R}^d \to \mathbb{R}^k$ and $B : \mathbb{R}^k \to \mathbb{R}$ are continuously differentiable ($C^1$) on open sets containing the paths

$$\{\overline{h} + t\Delta h : t \in [0,1]\}, \quad \{y(t) = Proj(\overline{h} + t\Delta h) : t \in [0,1]\}.$$

---

**Algorithm 1** Stereotype Cue Selection

---

**Require:** Pre-trained language model $M$, Templates $T_{adj}$, $T_{noun}$, Candidate cues $V_{adj}$ and $V_{noun}$, Demographic groups $D$, Entropy function $H(\cdot)$
**Ensure:** Top $k$ biased adjectives and nouns
 1: **function** COMPUTEENTROPIES($V, T$)
 2:     $E \leftarrow \{\}$
 3:     **for** $w \in V$ **do**
 4:         $p_{agg} \leftarrow 0$
 5:         **for** $t \in T$ **do**
 6:             $prompt \leftarrow$ Replace$(t, w)$
 7:             $p \leftarrow M(prompt)$
 8:             $p_{agg} \leftarrow p_{agg} + p/|T|$
 9:         **end for**
10:         $E \leftarrow E \cup H(p_{agg})$
11:     **end for**
12:     **return** $E$
13: **end function**
14: $E_{\text{adj}} \leftarrow$ COMPUTEENTROPIES($V_{\text{adj}}, T_{\text{adj}}$)
15: $E_{\text{noun}} \leftarrow$ COMPUTEENTROPIES($V_{\text{noun}}, T_{\text{noun}}$)
16: **Select** top $k$ cues from $E_{\text{adj}}$ and $E_{\text{noun}}$ based on lowest entropy

---

Hence $\nabla B$ is a continuous gradient field, and the line integral of $\nabla B$ is path-independent.

1. **Function Definitions**:

   - Model output function: $y = Proj(h)$, where $h \in \mathbb{R}^d$ denotes the input to the final hidden layer.
   - Bias function: $B : \mathbb{R}^k \to \mathbb{R}$, which takes the output distribution $y$ as input.
   - Composite function: $F(h) = B(Proj(h))$, mapping $h$ to the bias value.

2. **Modification Procedure**:

   - The modification set $S \subseteq \{1, \ldots, d\}$ (with cardinality $|S| = \beta d$) contains neurons with the highest positive attribution values.
   - Modified input:

$$h_j^{\text{mod}} = \begin{cases} C & j \in S \\ \overline{h}_j & j \notin S \end{cases} \tag{1}$$

   - Modification vector: $\Delta h = h^{\text{mod}} - \overline{h}$, with

$$\Delta h_j = \begin{cases} C - \overline{h}_j & j \in S \\ 0 & j \notin S \end{cases} \tag{2}$$

### A.6.2 STEP 1: EXACT EXPRESSION OF BIAS REDUCTION

By definition of the composite function $F$, the bias reduction is given by:

$$\Delta B = F(h^{\text{mod}}) - F(\overline{h}) \tag{3}$$

Using the integral form of the Mean Value Theorem along the path $\overline{h} \to h^{\text{mod}} = \overline{h} + \Delta h$, we have:

$$\Delta B = \int_0^1 \nabla F(\overline{h} + t\Delta h) \cdot \Delta h \, dt \tag{4}$$

Expanding the dot product:

$$\Delta B = \sum_{j=1}^d \Delta h_j \int_0^1 \frac{\partial F}{\partial h_j}(\overline{h} + t\Delta h) \, dt \tag{5}$$

Since $\Delta h_j = 0$ for $j \notin S$, this reduces to:

$$\Delta B = \sum_{j \in S}(C - \overline{h}_j) \int_0^1 \frac{\partial F}{\partial h_j}(\overline{h} + t\Delta h)\, dt \tag{6}$$

### A.6.3 STEP 2: EXACT EXPRESSION OF OUTPUT DISTRIBUTION VARIATION

For each component $y_i = Proj_i(h)$, the change is:

$$\Delta y_i = Proj_i(h^{\text{mod}}) - Proj_i(\overline{h}) \tag{7}$$

Applying the integral form of the Mean Value Theorem:

$$\Delta y_i = \int_0^1 \nabla Proj_i(\overline{h} + t\Delta h) \cdot \Delta h\, dt \tag{8}$$

Expanding:

$$\Delta y_i = \sum_{j=1}^d \Delta h_j \int_0^1 \frac{\partial Proj_i}{\partial h_j}(\overline{h} + t\Delta h)\, dt \tag{9}$$

Again, due to sparsity of $\Delta h$:

$$\Delta y_i = \sum_{j \in S}(C - \overline{h}_j) \int_0^1 \frac{\partial Proj_i}{\partial h_j}(\overline{h} + t\Delta h)\, dt \tag{10}$$

### A.6.4 STEP 3: RELATIONSHIP BETWEEN BIAS REDUCTION AND OUTPUT VARIATION

Using the chain rule on the composite function $F = B \circ Proj$, we denote the components of the projection function by $y_m = \text{Proj}_m$. Then

$$\frac{\partial F}{\partial h_j} = \sum_{m=1}^k \frac{\partial B}{\partial y_m}\frac{\partial y_m}{\partial h_j} = \sum_{m=1}^k \frac{\partial B}{\partial y_m}\frac{\partial Proj_m}{\partial h_j}. \tag{11}$$

Substituting (11) into (6):

$$\Delta B = \sum_{j \in S}(C - \overline{h}_j) \int_0^1 \left(\sum_{m=1}^k \frac{\partial B}{\partial y_m}(y(t)) \cdot \frac{\partial Proj_m}{\partial h_j}(\overline{h} + t\Delta h)\right) dt \tag{12}$$

with $y(t) = Proj(\overline{h} + t\Delta h)$. Interchanging the order of summation and integration:

$$\Delta B = \sum_{m=1}^k \int_0^1 \frac{\partial B}{\partial y_m}(y(t)) \cdot \left(\sum_{j \in S}(C - \overline{h}_j)\frac{\partial Proj_m}{\partial h_j}(\overline{h} + t\Delta h)\right) dt \tag{13}$$

Define the output-space velocity vector

$$G(t) := \frac{d}{dt}Proj(\overline{h} + t\Delta h) = \left[\sum_{j \in S}(C - \overline{h}_j)\frac{\partial Proj_m}{\partial h_j}(\overline{h} + t\Delta h)\right]_{m=1}^k . \tag{14}$$

Then (13) becomes

$$\Delta B = \int_0^1 \nabla B(y(t))^\top G(t)\, dt = \int_{y(0)}^{y(1)} \nabla B(y) \cdot dy. \tag{15}$$

Since $\nabla B$ is a conservative vector field, the line integral depends only on the endpoints $y(0), y(1)$.

### A.6.5 STEP 4: FINAL RELATIONSHIP

We can therefore replace the original curved path $y(t)$ with the straight-line path in output space:

$$u(s) = y(0) + s\Delta y, \quad s \in [0, 1], \tag{16}$$

where $\Delta y = y(1) - y(0)$. Thus,

$$\Delta B = \int_0^1 \nabla B(u(s))^\top \Delta y\, ds. \tag{17}$$

Define

$$\phi(s) := \nabla B(u(s))^\top \Delta y,$$

which is continuous on $[0, 1]$. By the Mean Value Theorem for integrals, there exists some $\theta \in [0, 1]$ such that

$$\Delta B = \phi(\theta) = \nabla B(y(0) + \theta\Delta y)^\top \Delta y. \tag{18}$$

**Final Result.**

$$\left| \Delta B \right| = \left| \nabla B(y(0) + \theta\Delta y)^\top \Delta y \right| \leqslant \|\nabla B(y(0) + \theta\Delta y)\| \cdot \|\Delta y\|, \quad \theta \in [0, 1]. \tag{19}$$

This completes the proof.

### A.7 COMPLETE EXPERIMENTS

#### A.7.1 COMPLETE RESULTS ACROSS DOMAINS ON THE BBQ DATASET

Table 8: Evaluation result across three demographic attributes on the BBQ dataset.

| Llama-3.1 | Gender | Nationality | Religion |
|---|---|---|---|
| | $\mathrm{Acc}_{amb}$ (%) ↑ / $\mathrm{Acc}_{dis}$ (%) ↑ / Average (%) ↑ | | |
| Base | 69.07 / 84.64 / 76.86 | 33.77 / 96.23 / 65.00 | 60.67 / 92.67 / 76.67 |
| Auto-Debias | 68.86 / 84.29 / 76.58 | 33.63 / 96.23 / 64.93 | 61.33 / 92.67 / 77.00 |
| Prefix Prompting | 87.86 / 79.43 / 83.65 | 59.87 / 94.81 / 77.34 | 77.67 / 84.00 / 80.84 |
| Self-Debiasing | 86.50 / 58.64 / 72.57 | 88.70 / 86.75 / 87.72 | 88.00 / 73.67 / 80.84 |
| DDP | 81.50 / 83.71 / 82.61 | 35.45 / 94.42 / 64.94 | 53.67 / 91.67 / 72.67 |
| IG$^2$ | 75.29 / 81.07 / 78.18 | 33.38 / 95.97 / 64.68 | 68.33 / 91.33 / 79.83 |
| FBA | 73.64 / 83.07 / 78.36 | 53.38 / 94.54 / 73.96 | 72.33 / 89.00 / 80.67 |
| BBA | 70.71 / 84.86 / 77.79 | 74.67 / 92.98 / 83.83 | 78.00 / 89.00 / 83.50 |

| Llama-3.2 | Gender | Nationality | Religion |
|---|---|---|---|
| | $\mathrm{Acc}_{amb}$ (%) ↑ / $\mathrm{Acc}_{dis}$ (%) ↑ / Average (%) ↑ | | |
| Base | 68.21 / 76.00 / 72.11 | 51.30 / 94.16 / 72.73 | 57.33 / 83.67 / 70.50 |
| Auto-Debias | 68.00 / 76.64 / 72.32 | 50.26 / 93.90 / 72.08 | 57.33 / 83.33 / 70.33 |
| Prefix Prompting | 86.71 / 62.57 / 74.64 | 78.05 / 85.19 / 81.62 | 83.67 / 59.33 / 71.50 |
| Self-Debiasing | 65.21 / 69.36 / 67.29 | 64.29 / 84.81 / 74.55 | 68.67 / 68.33 / 68.50 |
| DDP | 78.35 / 73.07 / 75.71 | 52.47 / 93.12 / 72.80 | 50.33 / 84.00 / 67.17 |
| IG$^2$ | 25.50 / 58.71 / 42.11 | 67.66 / 92.47 / 80.07 | 64.67 / 76.33 / 70.50 |
| FBA | 72.00 / 73.57 / 72.79 | 67.01 / 92.21 / 79.61 | 67.33 / 79.00 / 73.17 |
| BBA | 72.43 / 71.14 / 71.79 | 59.48 / 93.38 / 76.43 | 72.33 / 75.33 / 73.83 |

#### A.7.2 RESULTS ON THE BIAS-IN-BIOS DATASET

Bias-in-Bios (De-Arteaga et al., 2019) is a thirdperson biography dataset annotated by occupation and gender. We use LLMs to predict an individual's profession given their biography.

**Metric.** For the Bias-in-Bios dataset, we adopt the five evaluation metrics from (He et al., 2022), including: (1) $\mathrm{Acc}_{all}$, overall accuracy; (2) $\mathrm{Acc}_m$, accuracy on male-labeled instances; (3) $\mathrm{Acc}_f$, accuracy on female-labeled instances; (4) Gap-TPR, the difference in true positive rate (TPR) between male- and female-labeled instances; (5) RMS-TPR, the root-mean-square of the TPR gap

Table 9: Evaluation results on the `Bias-in-Bio` dataset.

| Llama-3.1 | $\text{Acc}_{all}$ | $\text{Acc}_m$ | $\text{Acc}_f$ | Gap-TPR | RMS-TPR |
|---|---|---|---|---|---|
| Base | 72.44 | 69.16 | 76.39 | 6.05 | 28.50 |
| Auto-Debias | 72.31 | 68.94 | 76.39 | 6.14 | 28.43 |
| Prefix Prompting | 73.72 | 69.83 | 78.41 | 7.30 | 28.85 |
| Self-Debiasing | 76.46 | 73.93 | 79.47 | 5.54 | 25.87 |
| DDP | 77.11 | 74.30 | 80.47 | 5.72 | 30.41 |
| $\text{IG}^2$ | 71.44 | 68.60 | 74.88 | 4.42 | 27.83 |
| FBA | 73.35 | 69.93 | 77.51 | 5.86 | 26.43 |
| BBA | 71.21 | 71.60 | 70.59 | **1.39** | **6.53** |
| **Llama-3.2** | $\text{Acc}_{all}$ | $\text{Acc}_m$ | $\text{Acc}_f$ | Gap-TPR | RMS-TPR |
| Base | 71.47 | 68.68 | 74.84 | 4.50 | 25.61 |
| Auto-Debias | 71.78 | 69.20 | 74.87 | 4.29 | 27.29 |
| Prefix Prompting | 70.47 | 67.43 | 74.12 | 5.38 | 30.88 |
| Self-Debiasing | 70.70 | 67.93 | 74.03 | 5.44 | 30.23 |
| DDP | 54.92 | 53.92 | 56.14 | 1.20 | 29.08 |
| $\text{IG}^2$ | 72.04 | 69.87 | 74.62 | 3.79 | 25.13 |
| FBA | 77.32 | 76.19 | 78.54 | 2.38 | 14.90 |
| BBA | 75.00 | 70.01 | 79.41 | **1.02** | **10.48** |

across all occupation classes. We selected a lightweight version of the `Bias-in-Bios` dataset[2] for testing. The experimental results on `Bias-in-Bios` are shown in Table 9.

We observe that BBA achieves a significantly stronger debiasing effect on `Bias-in-Bios` compared to all other methods (including FBA). Meanwhile, on Llama-3.2, both FBA and BBA improve all accuracy metrics.

### A.7.3 RESULTS ON THE BIAS-NLI DATASET

`Bias-NLI` (Dev et al., 2020) is an NLI dataset consisting of neutral sentence pairs. It is systematically constructed by populating sentence templates with a gendered word and an occupation word with a strong gender connotation (e.g., The woman ate a bagel; The nurse ate a bagel).

**Metrics.** For the `Bias-NLI` dataset, we evaluate large language models directly using prompt rather than performing classification with a fine-tuned BERT model as in (He et al., 2022). We compute the probabilities of the three labels (entailment, neutral, contradiction), denoted as $P_e$, $P_n$, and $P_c$. A higher value of $P_n$ indicates that the model is more fair.

Because the `Bias-NLI` dataset is exceptionally large, we selected the first 1,000 samples for testing. We find that Llama-3.2 is almost unable to perform correct linguistic reasoning on this dataset, as shown in the following Table 10:

Table 10: Evaluation results on the `Bias-NLI` dataset (Llama-3.2).

| Llama-3.2 | $P_e$ | $P_n$ | $P_c$ |
|---|---|---|---|
| base | 22.7 | 0.3 | 77.0 |

The results on Llama-3.2 show an abnormally low $P_n$, so we primarily focus on the results obtained with Llama-3.1 (Table 11).

Both FBA and BBA substantially increase $P_n$, and the accuracy of FBA is nearly 100%. This demonstrates that our method enables the model to correctly rule out stereotype-driven reasoning errors when inferring the relationship between sentences.

---

[2]`https://huggingface.co/datasets/LabHC/Bias_in_Bios_stratify`

Table 11: Evaluation results on the `Bias-NLI` dataset (Llama-3.1).

| Llama-3.1 | $P_e$ | $P_n$ | $P_c$ |
|---|---|---|---|
| base | 17.4 | 81.8 | 0.8 |
| Auto-Debias | 15.3 | 84.0 | 0.7 |
| Prefix Prompting | 15.1 | 84.3 | 0.6 |
| Self-Debiasing | 0.6 | 75.9 | 23.5 |
| DDP | 38.4 | 61.1 | 0.5 |
| IG$^2$ | 12.5 | 86.9 | 0.6 |
| FBA | 0.9 | **99.1** | 0.0 |
| BBA | 0.4 | 97.5 | 2.1 |

Table 12: Forward-IG and Backward-IG magnitude across layers for Llama-3.1 and Llama-3.2.

| Llama-3.1 | Gender | Nationality | Profession | Religion |
|---|---|---|---|---|
| Forward-IG (last layer) | $1.72 \times 10^{-3}$ | $4.49 \times 10^{-4}$ | $1.14 \times 10^{-2}$ | $3.81 \times 10^{-3}$ |
| Forward-IG (first layer) | $9.94 \times 10^{-8}$ | $8.14 \times 10^{-9}$ | $7.31 \times 10^{-9}$ | $1.71 \times 10^{-8}$ |
| Backward-IG (last layer) | $6.89 \times 10^{-5}$ | $3.57 \times 10^{-6}$ | $2.09 \times 10^{-5}$ | $4.25 \times 10^{-5}$ |
| Backward-IG (first layer) | $1.71 \times 10^{-7}$ | $2.85 \times 10^{-8}$ | $2.93 \times 10^{-8}$ | $1.82 \times 10^{-7}$ |
| **Llama-3.2** | **Gender** | **Nationality** | **Profession** | **Religion** |
| Forward-IG (last layer) | $9.03 \times 10^{-4}$ | $7.68 \times 10^{-4}$ | $3.10 \times 10^{-3}$ | $1.31 \times 10^{-3}$ |
| Forward-IG (first layer) | $9.97 \times 10^{-6}$ | $8.83 \times 10^{-7}$ | $6.09 \times 10^{-6}$ | $2.53 \times 10^{-5}$ |
| Backward-IG (last layer) | $7.61 \times 10^{-5}$ | $2.62 \times 10^{-6}$ | $1.10 \times 10^{-5}$ | $6.67 \times 10^{-5}$ |
| Backward-IG (first layer) | $5.82 \times 10^{-8}$ | $2.85 \times 10^{-8}$ | $2.47 \times 10^{-6}$ | $6.12 \times 10^{-6}$ |

Table 13: Evaluation results (`StereoSet`) for FBA and BBA across layers on Llama-3.1 and Llama-3.2.

| Llama-3.1 | Gender | Nationality | Profession | Religion |
|---|---|---|---|---|
| FBA (last layer) | 68.75/100/**62.50** | 64.88/97.09/**68.19** | 67.87/96.05/**61.73** | 51.35/93.67/**91.14** |
| FBA (first layer) | 75.59/99.22/48.44 | 64.96/97.30/**68.19** | 72.41/97.53/53.83 | 52.78/91.14/86.08 |
| BBA (last layer) | 69.84/98.44/**59.38** | 63.18/95.43/**70.27** | 71.58/95.56/**54.32** | 49.31/92.41/**91.14** |
| BBA (first layer) | 73.02/98.44/53.13 | 66.67/96.67/64.45 | 75.38/97.28/47.90 | 58.11/93.67/78.48 |
| **Llama-3.2** | **Gender** | **Nationality** | **Profession** | **Religion** |
| FBA (last layer) | 69.60/97.66/**59.38** | 58.52/95.22/**79.00** | 61.88/94.57/72.10 | 51.95/97.46/**93.67** |
| FBA (first layer) | 70.97/96.88/56.25 | 62.69/95.84/71.72 | 60.68/94.81/**74.56** | 59.74/97.47/78.48 |
| BBA (last layer) | 67.46/98.44/**64.06** | 62.93/96.47/71.52 | 62.21/96.05/**72.59** | 55.33/94.94/**88.61** |
| BBA (first layer) | 71.20/97.66/56.25 | 57.36/96.05/**81.91** | 64.87/96.30/67.65 | 56.00/94.93/83.54 |

### A.7.4 ANALYSIS OF LOWER-LAYER NEURON CONTRIBUTIONS

We compute the mean Forward-IG and Backward-IG values for the neurons in the first-layer hidden state and compare them with those from the final layer used in our experiments, as shown in Table 12. Most lower-layer neurons exhibit gradient signals that decay by more than two orders of magnitude, with the most severe cases diminishing by up to seven orders of magnitude. Such weakened gradient signals prevent the model from achieving optimal debiasing performance. We verify this by applying neuron-level modifications to lower layers on the `StereoSet` benchmark (Table 13).

### A.7.5 EXPERIMENTAL RESULTS IN MISTRAL-V0.3

Tables 14 and 15 present the performance of Mistral-v0.3 on the `StereoSet` and `WinoBias` datasets, respectively. On `StereoSet`, both FBA and BBA demonstrate certain effectiveness, particularly in the domains of profession and religion, where FBA achieves notably strong performance. On `WinoBias`, although we did not achieve the optimal Gap, our approach still significantly outperforms Self-Debiasing, which causes a sharp increase in $P_{other}$. Overall, BBA attains the second-best performance.

Table 14: Evaluation results across four demographic domains on the `StereoSet` dataset.

| Mistral-v0.3 | Gender | Nationality | Profession | Religion |
|---|---|---|---|---|
| | SS (%) → 50% / LMS (%) ↑ / ICAT (%) ↑ | | | |
| Base | 72.36 / 96.09 / 53.12 | 60.00 / 92.52 / 74.01 | 69.39 / 93.58 / 57.28 | 60.81 / 93.67 / 73.42 |
| Auto-Debias | 77.12 / 92.19 / 42.19 | 61.50 / 91.27 / 70.27 | 70.08 / 91.60 / 54.81 | 60.00 / 94.94 / 75.95 |
| Prefix Prompting | 72.27 / 92.97 / 51.56 | 55.10 / 91.68 / **82.33** | 64.27 / 92.59 / 66.17 | 52.86 / 88.61 / **83.54** |
| Self-Debiasing | 42.70 / 69.53 / 59.37 | 48.09 / 70.89 / 68.19 | 56.51 / 72.10 / 62.72 | 55.93 / 74.68 / 65.82 |
| DDP | 66.39 / 95.31 / **64.06** | 58.65 / 92.52 / 76.50 | 66.93 / 94.07 / 62.22 | 63.24 / 86.08 / 63.29 |
| IG$^2$ | 76.03 / 94.53 / 45.31 | 45.96 / 66.94 / 61.54 | 51.81 / 68.15 / 65.68 | 51.79 / 70.89 / 68.35 |
| FBA | 68.33 / 93.75 / 59.38 | 55.11 / 89.40 / 80.25 | 65.80 / 94.57 / 64.69 | 55.40 / 93.67 / **83.54** |
| BBA | 68.33 / 93.75 / 59.38 | 54.19 / 84.41 / 77.34 | 59.08 / 85.68 / **70.12** | 54.93 / 89.87 / 81.01 |

Table 15: Evaluation results on the `WinoBias` dataset.

| | Mistral-v0.3 | | | |
|---|---|---|---|---|
| | $P_{stereo}$ | $P_{anti}$ | $P_{other}$ ↓ | $Gap$ ↓ |
| Base | 61.11 | 38.64 | 0.25 | 22.47 |
| Auto-Debias | 56.69 | 41.16 | 2.15 | 15.53 |
| Prefix Prompting | 71.21 | 28.79 | 0.00 | 42.42 |
| Self-Debiasing | 49.49 | 43.43 | 7.08 | 6.06 |
| IG$^2$ | 50.76 | 48.48 | 0.76 | **2.28** |
| FBA | 58.59 | 40.66 | 0.75 | 17.93 |
| BBA | 55.81 | 43.43 | 0.76 | 12.38 |

### A.7.6    EXTRA ABLATION RESULTS ON STEREOSET

Some ablation results have already been presented in the main text. Figures ( 3,4,5,6,7,8,9) report the ablation results of all models on StereoSet. In most cases, our two attribution methods achieve a simultaneous improvement in both SS and LMS. In certain cases, when LMS values are comparable, our methods yield SS scores closer to 50%.

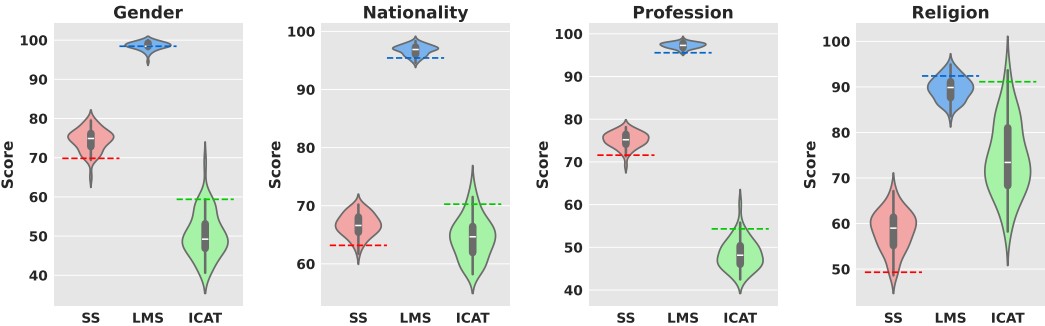

Figure 3: Ablation results of Llama-3.1 on StereoSet: BBA w/o attribution.

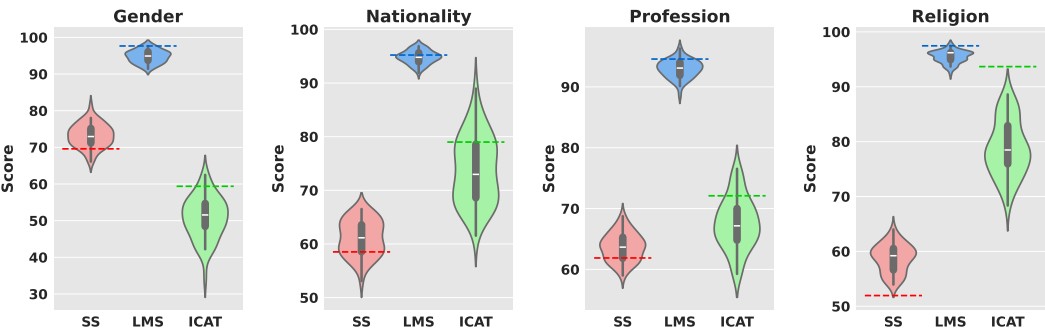

Figure 4: Ablation results of Llama-3.2 on StereoSet: FBA w/o attribution.

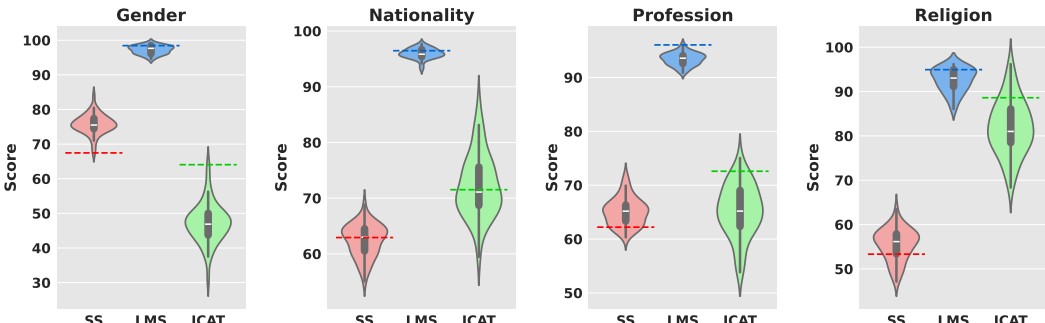

Figure 5: Ablation results of Llama-3.2 on StereoSet: BBA w/o attribution.

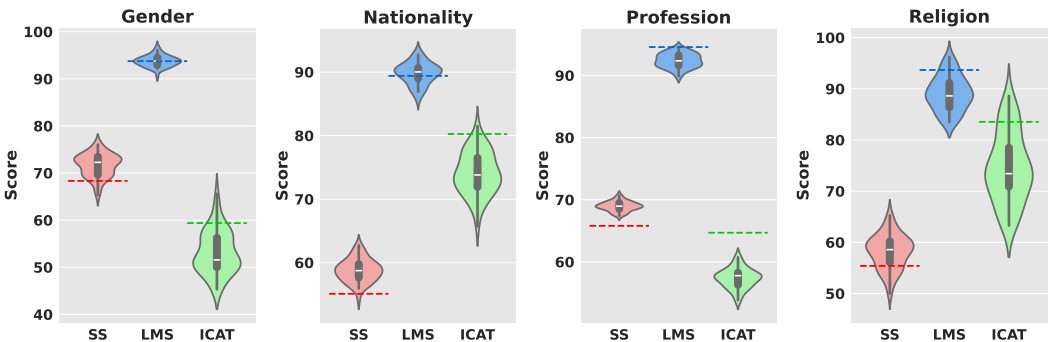

Figure 6: Ablation results of Mistral-v0.3 on `StereoSet`: FBA w/o attribution.

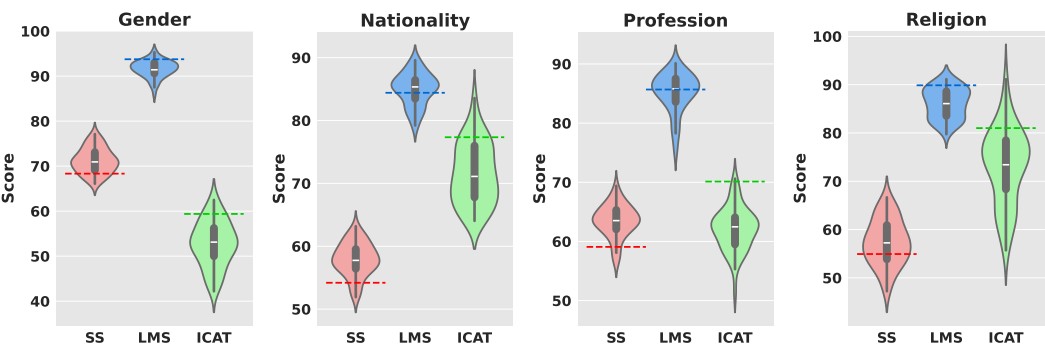

Figure 7: Ablation results of Mistral-v0.3 on `StereoSet`: BBA w/o attribution.

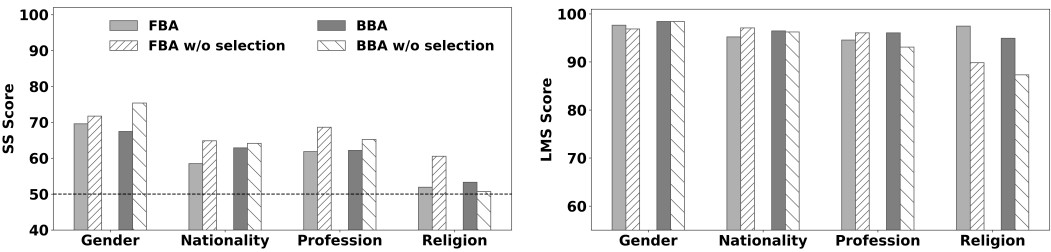

Figure 8: Ablation results of Llama-3.2 on `StereoSet`: w/o selection.

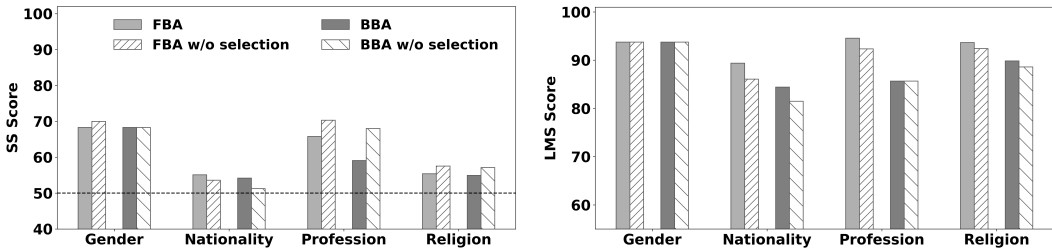

Figure 9: Ablation results of Mistral-v0.3 on `StereoSet`: w/o selection.

### A.7.7   ABLATION RESULTS ON WINOBIAS

Tables (16,17,18) report the ablation results of all models on WinoBias. When our attribution strategy is replaced with randomly selected neurons, the Gap value rises sharply, indicating that the model bias is not alleviated at all. Moreover, when stereotype cue selection is removed, a decline in debiasing performance is observed across all settings except for the FBA method on Llama-3.2.

Table 16: Ablation results of Llama-3.1 on WinoBias.

| | Llama-3.1 | | | |
|---|---|---|---|---|
| | $P_{stereo}$ | $P_{anti}$ | $P_{other} \downarrow$ | $Gap \downarrow$ |
| FBA | 52.53 | 47.47 | 0.00 | 5.06 |
| FBA w/o attribution | 62.4 | 36.75 | 0.85 | 25.65 |
| FBA w/o seletion | 57.58 | 42.42 | 0.00 | 15.16 |
| BBA | 50.51 | 49.49 | 0.00 | 1.02 |
| BBA w/o attribution | 59.85 | 40.15 | 0.00 | 19.70 |
| BBA w/o seletion | 49.49 | 50.51 | 0.00 | 1.02 |

Table 17: Ablation results of Llama-3.2 on WinoBias.

| | Llama-3.2 | | | |
|---|---|---|---|---|
| | $P_{stereo}$ | $P_{anti}$ | $P_{other} \downarrow$ | $Gap \downarrow$ |
| FBA | 59.34 | 40.66 | 0.00 | 18.68 |
| FBA w/o attribution | 74.75 | 21.16 | 4.09 | 53.59 |
| FBA w/o seletion | 57.83 | 42.17 | 0.00 | 15.66 |
| BBA | 51.52 | 48.48 | 0.00 | 3.04 |
| BBA w/o attribution | 75.72 | 19.15 | 5.13 | 56.57 |
| BBA w/o seletion | 55.30 | 44.70 | 0.00 | 10.60 |

Table 18: Ablation results of Mistral-v0.3 on WinoBias.

| | Mistral-v0.3 | | | |
|---|---|---|---|---|
| | $P_{stereo}$ | $P_{anti}$ | $P_{other} \downarrow$ | $Gap \downarrow$ |
| FBA | 58.59 | 40.66 | 0.75 | 17.93 |
| FBA w/o attribution | 64.48 | 31.20 | 4.32 | 33.28 |
| FBA w/o seletion | 57.32 | 39.14 | 3.54 | 18.18 |
| BBA | 55.81 | 43.43 | 0.76 | 12.38 |
| BBA w/o attribution | 61.50 | 37.66 | 0.84 | 23.84 |
| BBA w/o seletion | 62.88 | 35.10 | 2.02 | 27.78 |

A.7.8 HYPERPARAMETER SETTINGS AND SENSITIVITY ANALYSIS

Since our method does not rely on a training set, we split `StereoSet` and `WinoBias` into validation and test sets at a 1:1 ratio. However, due to the limited number of samples in the religion domain of `StereoSet`, this domain could not be partitioned. For the approximate computation of Forward-IG and Backward-IG, we set the number of approximation steps to $n_{step} = 20$.

In the process of modifying neuron activations, two parameters are involved: the modification ratio $\beta$ and the constant value $C$ to which the activations are set. Arbitrarily setting the constant $C$ may exacerbate bias, and therefore we perform a grid search over the parameters. Specifically, we set the search range of $\beta$ to $[0.1, 0.2, 0.3, 0.4]$, and the range of $C$ to $[-2, -1, 0, 1, 2]$. Tables (19,20,21,22,23,24,25,26) present the grid search results of FBA and BBA on the Llama-3.1 model with `StereoSet`. The gray cells indicate the parameters ultimately adopted. We observe that larger absolute values of $\beta$ and $C$ tend to cause a more severe degradation of the model's language modeling ability (i.e., LMS), while simultaneously yielding stronger debiasing effects. This phenomenon is consistent with our previously established Theorem 1.

Table 19: Hyperparameter search of the FBA method on the gender domain (SS, LMS).

| $\beta$ \ $C$ | -2 | -1 | 0 | 1 | 2 |
|---|---|---|---|---|---|
| 0.1 | (68.00, 97.66) | (67.46, 98.44) | (73.44, 100.0) | (73.23, 99.22) | (68.75, 100.0) |
| 0.2 | (59.32, 92.19) | (68.80, 97.66) | (72.44, 99.22) | (74.19, 96.88) | (71.43, 92.97) |
| 0.3 | (58.18, 85.94) | (66.40, 97.66) | (78.05, 96.09) | (78.57, 98.44) | (60.91, 85.94) |
| 0.4 | (46.15, 71.09) | (67.74, 96.88) | (71.43, 98.44) | (68.85, 95.31) | (47.78, 70.31) |

Table 20: Hyperparameter search of the FBA method on the nationality domain (SS, LMS).

| $\beta$ \ $C$ | -2 | -1 | 0 | 1 | 2 |
|---|---|---|---|---|---|
| 0.1 | (69.25, 96.67) | (68.51, 97.71) | (68.86, 98.13) | (69.02, 97.30) | (65.17, 97.30) |
| 0.2 | (68.20, 94.80) | (69.72, 97.51) | (69.43, 97.92) | (64.88, 97.09) | (58.76, 93.76) |
| 0.3 | (60.43, 86.69) | (68.13, 94.59) | (67.95, 97.30) | (64.78, 95.63) | (59.06, 88.36) |
| 0.4 | (52.52, 70.06) | (68.58, 93.97) | (68.71, 98.34) | (62.23, 95.22) | (50.43, 72.56) |

Table 21: Hyperparameter search of the FBA method on the profession domain (SS, LMS).

| $\beta$ \ $C$ | -2 | -1 | 0 | 1 | 2 |
|---|---|---|---|---|---|
| 0.1 | (70.36, 95.80) | (75.96, 96.54) | (76.20, 97.53) | (73.05, 98.02) | (73.55, 98.02) |
| 0.2 | (68.60, 93.58) | (73.08, 96.30) | (75.76, 97.78) | (74.11, 97.28) | (69.41, 92.84) |
| 0.3 | (60.34, 87.16) | (67.87, 96.05) | (72.73, 97.78) | (71.25, 97.04) | (66.48, 88.40) |
| 0.4 | (55.31, 76.79) | (64.17, 92.35) | (72.91, 97.53) | (68.24, 94.07) | (60.47, 74.32) |

Table 22: Hyperparameter search of the FBA method on the religion domain (SS, LMS).

| $\beta$ \ $C$ | -2 | -1 | 0 | 1 | 2 |
|---|---|---|---|---|---|
| 0.1 | (58.90, 92.41) | (61.64, 92.41) | (63.01, 92.41) | (59.15, 89.87) | (59.72, 91.14) |
| 0.2 | (51.35, 93.67) | (54.93, 89.87) | (65.28, 91.14) | (63.01, 92.41) | (63.38, 89.87) |
| 0.3 | (59.42, 87.34) | (58.11, 93.67) | (55.41, 93.67) | (57.75, 89.87) | (58.46, 82.28) |
| 0.4 | (56.90, 73.42) | (62.50, 81.01) | (62.50, 91.14) | (61.11, 91.14) | (56.67, 75.95) |

Table 23: Hyperparameter search of the BBA method on the gender domain (SS, LMS).

| $\beta$ \ $C$ | -2 | -1 | 0 | 1 | 2 |
|---|---|---|---|---|---|
| 0.1 | (73.02, 98.44) | (75.59, 99.22) | (75.59, 99.22) | (76.19, 98.44) | (74.59, 95.31) |
| 0.2 | (63.56, 92.19) | (69.84, 98.44) | (73.23, 99.22) | (73.81, 98.44) | (74.36, 91.41) |
| 0.3 | (58.82, 79.69) | (62.50, 93.75) | (71.20, 97.66) | (76.61, 96.88) | (69.81, 82.81) |
| 0.4 | (51.16, 67.19) | (61.95, 88.28) | (70.16, 96.88) | (75.63, 92.97) | (52.75, 71.09) |

Table 24: Hyperparameter search of the BBA method on the nationality domain (SS, LMS).

| $\beta$ \ $C$ | -2 | -1 | 0 | 1 | 2 |
|---|---|---|---|---|---|
| 0.1 | (65.43, 95.01) | (66.88, 97.30) | (66.45, 97.30) | (66.60, 97.71) | (64.96, 97.30) |
| 0.2 | (60.92, 90.44) | (63.18, 95.43) | (65.45, 96.88) | (65.58, 95.43) | (63.35, 91.89) |
| 0.3 | (58.42, 81.50) | (65.19, 93.76) | (65.13, 94.80) | (64.33, 92.10) | (62.47, 82.54) |
| 0.4 | (49.12, 70.69) | (60.68, 85.65) | (61.28, 91.27) | (62.53, 85.45) | (56.73, 72.56) |

Table 25: Hyperparameter search of the BBA method on the profession domain (SS, LMS).

| $\beta$ \ $C$ | -2 | -1 | 0 | 1 | 2 |
|---|---|---|---|---|---|
| 0.1 | (73.91, 96.54) | (77.22, 97.53) | (75.76, 97.78) | (74.18, 97.53) | (72.89, 96.54) |
| 0.2 | (69.57, 90.86) | (74.17, 96.54) | (75.26, 96.79) | (72.42, 95.80) | (66.94, 91.11) |
| 0.3 | (65.88, 83.21) | (71.54, 94.57) | (71.58, 95.56) | (73.28, 93.33) | (62.54, 81.73) |
| 0.4 | (56.69, 70.12) | (69.36, 88.64) | (71.17, 95.06) | (66.00, 86.42) | (54.49, 74.32) |

Table 26: Hyperparameter search of the BBA method on the religion domain (SS, LMS).

| $\beta$ \ $C$ | -2 | -1 | 0 | 1 | 2 |
|---|---|---|---|---|---|
| 0.1 | (62.50, 91.14) | (61.11, 91.14) | (60.81, 93.67) | (59.46, 93.67) | (56.94, 91.14) |
| 0.2 | (59.42, 87.34) | (62.16, 93.67) | (57.53, 92.41) | (58.90, 92.41) | (49.32, 92.41) |
| 0.3 | (64.52, 78.48) | (63.89, 91.14) | (58.90, 92.41) | (54.79, 92.41) | (49.25, 84.81) |
| 0.4 | (49.12, 72.15) | (53.62, 87.34) | (56.16, 92.41) | (52.17, 87.34) | (49.12, 72.15) |

## A.8 Complete Templates for Two Types of Stereotype Cues

| Category | Template Examples |
|---|---|
| Adjective | The [Demographic_Attribute] of this [Stereotype_Adjective] person is [Demographic_Group]. |
| | This [Stereotype_Adjective] person belongs to the [Demographic_Attribute] of [Demographic_Group]. |
| | This person who is [Stereotype_Adjective] is of the [Demographic_Attribute] of [Demographic_Group]. |
| | This person who is [Stereotype_Adjective] is of the [Demographic_Group] [Demographic_Attribute]. |
| | This [Stereotype_Adjective] person is in the [Demographic_Attribute] of [Demographic_Group]. |
| | This [Stereotype_Adjective] person is in the [Demographic_Group] [Demographic_Attribute]. |
| | The [Stereotype_Adjective] person's [Demographic_Attribute] is identified as [Demographic_Group]. |
| | This [Stereotype_Adjective] person's [Demographic_Attribute] is [Demographic_Group]. |
| | The [Demographic_Attribute] of this person who is [Stereotype_Adjective] is identified as [Demographic_Group]. |
| | This [Stereotype_Adjective] person identifies as [Demographic_Group] in terms of [Demographic_Attribute]. |
| | This person who is [Stereotype_Adjective] identifies with the [Demographic_Group] [Demographic_Attribute]. |
| | In terms of [Demographic_Attribute], this [Stereotype_Adjective] person is identified as [Demographic_Group]. |
| | The [Demographic_Attribute] identification of this person who is [Stereotype_Adjective] is [Demographic_Group]. |
| | These [Stereotype_Adjective] people associate themselves with the [Demographic_Group] [Demographic_Attribute]. |
| | In terms of [Demographic_Attribute], these [Stereotype_Adjective] people identify themselves as [Demographic_Group]. |
| | These [Stereotype_Adjective] people identify themselves as [Demographic_Group] in relation to [Demographic_Attribute]. |
| | These people who are [Stereotype_Adjective] identify their [Demographic_Attribute] as [Demographic_Group]. |
| Noun | The [Demographic_Attribute] of this [Stereotype_Noun] is [Demographic_Group]. |
| | This [Stereotype_Noun] belongs to the [Demographic_Attribute] of [Demographic_Group]. |
| | This person, who is a [Stereotype_Noun], is of the [Demographic_Attribute] of [Demographic_Group]. |
| | This [Stereotype_Noun] is of the [Demographic_Group] [Demographic_Attribute]. |
| | This [Stereotype_Noun] is in the [Demographic_Attribute] of [Demographic_Group]. |
| | This [Stereotype_Noun] is in the [Demographic_Group] [Demographic_Attribute]. |
| | The [Demographic_Attribute] of the [Stereotype_Noun] is identified as [Demographic_Group]. |
| | This [Stereotype_Noun]'s [Demographic_Attribute] is [Demographic_Group]. |
| | The [Demographic_Attribute] of this person who is [Stereotype_Noun] is identified as [Demographic_Group]. |
| | This [Stereotype_Noun] identifies as [Demographic_Group] in terms of [Demographic_Attribute]. |
| | This person who is a [Stereotype_Noun] identifies with the [Demographic_Group] [Demographic_Attribute]. |
| | In terms of [Demographic_Attribute], this [Stereotype_Noun] is identified as [Demographic_Group]. |
| | The [Demographic_Attribute] identification of this [Stereotype_Noun] is [Demographic_Group]. |
| | These [Stereotype_Noun]s associate themselves with the [Demographic_Group] [Demographic_Attribute]. |
| | In terms of [Demographic_Attribute], these [Stereotype_Noun]s identify themselves as [Demographic_Group]. |
| | These [Stereotype_Noun]s identify themselves as [Demographic_Group] in relation to [Demographic_Attribute]. |
| | These people who are [Stereotype_Noun]s identify their [Demographic_Attribute] as [Demographic_Group]. |

## A.9    DEMOGRAPHIC GROUPS FOR ALL DEMOGRAPHIC ATTRIBUTES

| Demographic Attribute | Demographic Groups |
| --- | --- |
| Gender | male, female |
| Nationality | Russian, Ukrainian, Ethiopia, Afghanistan, Yemen, Somalia, Saudi Arabian, Italy, Crimean, Hispanic, Arab, Iraq, Syria, Eritrean, Japanese, Jordan, Norweigan, Ghanaian, Sierra Leon, Eriteria, Bengali, Vietnam, Iranian, Lebanon, Ecuador, Spain, Cameroon, Columbian, Morocco, Bangladesh, African, Norway, Britain, Ethiopian, Cape Verde, Persian people |
| Profession | football player, manager, prisoner, entrepreneur, prosecutor, mathematician, commander, chess player, musician, software developer, butcher, nurse, psychologist, mover, physicist, producer, bartender, researcher, politician, tailor, chemist, engineer, historian, plumber, civil servant, guitarist, performing artist, delivery man, policeman, assistant |
| Religion | Christian, Muslim, Buddhist, Hindus |

## A.10    PROMPTS FOR CONSTRUCTING QUESTIONS

---

**Prompts for FBA**

Based on the context, fill in the blank using the candidate options. Only answer with ONE word from the options without any other words.

Context: {question}

*// Template sentences in which [Demographic_Group] should be filled*

Options: {options}

*// Candidate demographic groups for filling the blank*

---

**Prompts for BBA**

According to the demographic attribute of the person, fill in the blank in the sentence using the following options.

Sentence: {question}

*// Template sentences in which [Stereotype_Adjective] or [Stereotype_Noun] should be filled*

Options: {options}

*// Candidate stereotype words for filling the blank*

Only provide a single word from the options, nothing else.

---

