# OpenReview forum: "Bi-directional Bias Attribution: Debiasing Large Language Models without Modifying Prompts"
_ICLR.cc/2026/Conference — ICLR 2026 Poster_

### Official Review · Reviewer_Ydo8 · 2025-10-21

**Soundness:** 3
**Presentation:** 3
**Contribution:** 3
**Rating:** 4
**Confidence:** 3

**Summary:**

This paper proposes a novel bias mitigation method for large language models (LLMs) based on bi-directional bias attribution, which modifies neuron weights directly rather than relying on fine-tuning or prompt modification. The approach first identifies stereotype-inducing words, computes forward and backward attribution scores to locate neurons that strongly mediate biased associations, and then adjusts those neurons’ outgoing weights to neutralize bias effects. The method is efficient and interpretable, making it attractive for bias correction without retraining. Experiments across standard bias benchmarks demonstrate meaningful bias reduction while largely preserving model quality.

I think the main contribution lies in connecting interpretability techniques with bias mitigation, creating a new approach to fairness interventions. The reliance on manually identified stereotype words make the approach somewhat limiting and the theoretical basis for weight adjustment and its generalizability beyond the tested contexts remain unclear.

**Strengths:**

Novel combination of interpretability for bias mitigation:
The paper presents a creative and technically meaningful idea: using bi-directional attribution to identify and correct bias-related neurons without fine-tuning or prompt engineering. While attribution-based interpretability methods are well known, directly leveraging them for bias mitigation is a fresh and promising direction that bridges the two important subfields.

Training-free, efficient, and potentially generalizable method:
The approach modifies existing model weights through targeted interventions rather than retraining, making it computationally light and practically attractive. This could be particularly useful for post-deployment fairness adjustments, where retraining is infeasible.

Mechanistic interpretability grounded framework:
By analyzing neuron activations through forward and backward attributions, the method provides a more interpretable view of where and how biases emerge inside the model. Even if attribution faithfulness is imperfect, the attempt to connect bias to internal mechanisms rather than surface outputs is conceptually strong.

Solid empirical evaluation:
The experiments are systematic, showing consistent bias reduction across multiple datasets while maintaining generation quality. The results demonstrate the feasibility of neuron-level interventions and position the method as a practical proof of concept.

**Weaknesses:**

Fragile theoretical grounding and interpretability assumptions:
The method assumes that gradient-based attributions faithfully capture causal influence of neurons on biased behavior, but this assumption is only weakly supported. Attribution methods in large transformers are known to be noisy, context-sensitive, and often misaligned with causal importance. Without rigorous validation, it remains unclear whether the identified “bias neurons” are genuinely responsible for the effects being mitigated. In terms of interpretability, there is little empirical evidence that the neuron-level adjustments are truly interpretable in practice, for example, by visualizing which features are changed or explaining their semantic role.

Dependence on predefined stereotype word lists:
The identification of bias-inducing neurons relies on manually or heuristically selected “stereotype words.” This injects subjective bias into the pipeline and may limit generalizability to less well-defined attributes or intersectional biases. The reliance on such lexicons also makes the approach difficult to scale beyond English or to subtle forms of representational bias.

Missing related work section:
The paper lacks a thorough review of prior interpretability-based bias or neuron-level editing approaches. The absence of this context makes it harder to evaluate the originality and significance of the proposed method.

Limited analysis of generalization and stability:
The evaluation focuses on standard bias benchmarks and does not examine whether the neuron edits generalize to other models, domains, or unseen contexts. There is also no robustness analysis to test stability across different random seeds or attribution thresholds. Given that attribution methods can vary across runs, this omission weakens confidence in reproducibility.

**Questions:**

The attribution and interpretability portion:
How confident are you that the forward and backward attribution signals identify neurons with genuine causal influence on biased behavior rather than merely correlated activation patterns?
Have you compared your neuron selection against causal intervention baselines (e.g., activation patching, representation ablation) to confirm that the edited neurons are indeed responsible for the observed bias?
Can you provide qualitative evidence (e.g., neuron visualizations, activation maps) demonstrating that your edits are interpretable and consistent across examples?

Reliance on stereotype word lists:
Do you expect the method to generalize to settings where biased cues are more subtle, implicit, or multimodal?
How does the method behave on other social dimensions beyond those represented in your word lists (e.g., intersectional or cultural biases)?

---

> ### Author Response · Authors · 2025-11-22
> **Official Comment to Reviewer Ydo8 (Part I)**
>
> Dear Reviewer Ydo8, thank you very much for your review and questions. Here are our responses to your comments.
>
> ---
> ## W1&Q1: Theoretical grounding and attribution assumptions
> IG² has demonstrated that, in masked language models, certain neurons are prone to inducing biased model behaviors. Our empirical findings show that attribution-identified neurons behave analogously to the bias neurons discovered in masked LMs. Specifically, we validate this hypothesis from the following perspectives.
> ### 1. Direct validation via activation-level intervention
> After ranking neurons with Forward-IG/Backward-IG, we _intervene_ on their activations at the projection layer and measure the resulting causal effect on outputs.
> - If the identified neurons are not causally responsible, intervening on them would produce performance collapse or no bias reduction.
> - However, our controlled interventions (where only the top-N neurons are fixed) consistently reduce bias without harming LMS scores (Table 5,6,7), demonstrating causal relevance.
> ### 2. Ablation against random intervention
> We performed **50** random-neuron intervention trials for each domain (Figure 2a, Appendix A.7.4), and found:
> - Random interventions hardly reduce bias and often degrade LMS heavily.
> - FBA/BBA consistently achieves **lower bias + higher LMS**.
> This is direct causal evidence that attribution-selected neurons differ fundamentally from correlated-but-irrelevant neurons.
> ### 3. Theoretical justification
> Theorem 1 links bias reduction $\Delta B$ to output variation $\Delta y$ induced by neuron intervention.
> Modifying a large number of neurons can produce a large $\Delta y$, and consequently a large $\Delta B$. However, our goal is to use the rankings from Forward-IG and Backward-IG to modify only a small subset of neurons while ensuring that the quantity $\left\| \nabla B\left( y _ 0 + \theta \Delta y \right)\right\|$ remains large. Therefore, the causal effect of neurons on bias should be understood as a **relative** notion: certain neurons exhibit stronger causal connections to bias compared with the remaining neurons.

---

> ### Author Response · Authors · 2025-11-22
> **Official Comment to Reviewer Ydo8 (Part II)**
>
> ### 4. Visualization
> For feature-level attribution methods such as IG, visualizations are intuitive and easy for humans to understand. For example, in an image classification task, we can highlight the features with high IG values and visually inspect whether the identified features align with the contours of the target object, thereby effectively demonstrating the attribution results. However, the settings of Forward-IG, Backward-IG, and IG² are similar in that attribution is performed at the **neuron level**, where individual neurons do not carry explicit semantic meaning. Thus, IG-style visualizations are not applicable.
>
> To help illustrate the debiasing effects of FBA and BBA, we conducted an additional visualization experiment on WinoBias, as shown in the newly added Figure 10 in Appendix A.7.7. We divide the identified bias-related neurons into ten equal groups and gradually increase the proportion of modified neurons. We then observe how the model’s preference between stereotypical and anti-stereotypical options changes accordingly, as shown below.
>
> **Table 1. Effects of Neuron Modification Ratio Using FBA on Llama 3.2.**
>
> | Ratio | $P _ {stereo}$ (%) | $P _ {anti}$ (%) | $P _ {other}$ (%) | $Gap$ (%) |
> | :---- | :--------------: | :------------: | :-------------: | :-------: |
> | 0.1   |      95.96       |     4.04       |      0.00       |   91.92   |
> | 0.2   |      96.21       |     3.79       |      0.00       |   92.42   |
> | 0.3   |      95.20       |     4.80       |      0.00       |   90.40   |
> | 0.4   |      93.43       |     6.57       |      0.00       |   86.87   |
> | 0.5   |      93.18       |     6.82       |      0.00       |   86.36   |
> | 0.6   |      93.18       |     6.82       |      0.00       |   86.36   |
> | 0.7   |      93.18       |     6.82       |      0.00       |   86.36   |
> | 0.8   |      92.93       |     7.07       |      0.00       |   85.86   |
> | 0.9   |      91.41       |     8.33       |      0.25       |   83.08   |
> | 1.0   |      61.62       |     38.13      |      0.25       |   23.48   |
>
> **Table 2. Effects of Neuron Modification Ratio Using BBA on Llama 3.2.**
>
> | Ratio | $P _ {stereo}$ (%) | $P _ {anti}$ (%) | $P _ {other}$ (%) | $Gap$ (%) |
> | :---- | :--------------: | :------------: | :-------------: | :-------: |
> | 0.1   |      92.42       |     7.58       |      0.00       |   84.85   |
> | 0.2   |      78.79       |     21.21      |      0.00       |   57.58   |
> | 0.3   |      80.56       |     19.44      |      0.00       |   61.11   |
> | 0.4   |      62.63       |     37.37      |      0.00       |   25.25   |
> | 0.5   |      65.66       |     34.34      |      0.00       |   31.31   |
> | 0.6   |      77.02       |     22.98      |      0.00       |   54.04   |
> | 0.7   |      73.74       |     26.26      |      0.00       |   47.47   |
> | 0.8   |      78.54       |     21.46      |      0.00       |   57.07   |
> | 0.9   |      63.38       |     36.62      |      0.00       |   26.77   |
> | 1.0   |      51.52       |     48.48      |      0.00       |    3.03   |
>
> We observe that as the number of modified neurons increases, the overall debiasing effect exhibits a clear downward trend, with incremental changes eventually leading to a qualitative shift.
> ### 5. Overall answers to Q1
> - **Q1-1. How confident are you the forward/backward signals identify causal neurons?**
>
> 	Yes, we are confident because editing these neurons changes bias, and editing other neurons does not, this is precisely the definition of causal influence. Ablations and IG² baseline comparison strongly support causal relation.
>
> - **Q1-2. Any comparison to causal methods?**
>
> 	Yes. IG² and random intervention is baseline attribution methods.  Our method outperforms IG² in debiasing and avoids LMS collapse.
>
> - **Q1-3. Any qualitative visualizations?**
>
> 	Since individual neurons do not carry explicit semantic meaning, we conducted an additional visualization experiment on WinoBias in Fig 10.

---

> ### Author Response · Authors · 2025-11-22
> **Official Comment to Reviewer Ydo8 (Part III)**
>
> ## W2&Q2: Dependence on predefined stereotype word lists
> We respectfully clarify that our stereotype cue selection is model-specific and entropy-based, not manually curated.
> ### 1. Our cue selection is automatic
> We start from a pool of adjectives/nouns and then filter them solely based on the model’s entropy response:
> - Lower entropy $\Rightarrow$ stronger model-specific bias activation.
>     This ensures cues match _the model’s internal representation_, not human intuitions.
> - Human stereotypes $\ne$ model stereotypes.
> An important fact is that cues humans think are “stereotypes” are often not the ones triggering the model. This is consistent with Auto-Debias [1]: many cue words look meaningless to humans, yet strongly activate model biases. Therefore, manual selection cannot serve our goal.
> ### 2. Ablation of cue selection
> Removing cue selection ("w/o selection", Fig. 2b, Appendix A.7.4 and Appendix A.7.5) leads to:
> - SS scores farther from 50% (i.e., more bias)
> -  Slightly lower LMS
> This empirically proves that entropy-selected cues indeed capture the model’s internal stereotype triggers.
>
> ### 3. Direct experimental validation of cue selection
> In our response to Reviewer FZkY’s W1, we noted that we added a direct validation of the selected stereotype cues. We define male-associated terms \($w _ m$ = ["male", "man"] \),  and female-associated terms \($w _ f$ = ["female", "woman"] \). For the top 5 selected cues, we compute their average cosine similarity to each vocabulary ($\mathrm{Sim} _ m$ and $\mathrm{Sim} _ f$) and the average absolute difference ($\mathrm{Diff}$). We perform the same computations for all candidate terms as a comparison.
>
> | Llama-3.1 (Adjective) |   $\mathrm{Sim} _ m$ (%)   |   $\mathrm{Sim} _ f$ (%)   |   $\mathrm{Diff}$ (%)   | Llama-3.2 (Adjective) |   $\mathrm{Sim} _ m$ (%)   |   $\mathrm{Sim} _ f$ (%)   |   $\mathrm{Diff}$ (%)   |
> | --------------------- | :----------------------: | :----------------------: | :---------------------: | --------------------- | :----------------------: | :----------------------: | :---------------------: |
> | Top-5 Selected Cues   |           8.92           |           4.60           |          4.32           | Top-5 Selected Cues   |           6.04           |           7.21           |          6.44           |
> | All Candidates        |           6.37           |           4.68           |          2.79           | All Candidates        |           7.76           |          11.07           |          6.11           |
> | **Llama-3.1 (Noun)**  | **$\mathrm{Sim} _ m$ (%)** | **$\mathrm{Sim} _ f$ (%)** | **$\mathrm{Diff}$ (%)** | **Llama-3.2 (Noun)**  | **$\mathrm{Sim} _ m$ (%)** | **$\mathrm{Sim} _ f$ (%)** | **$\mathrm{Diff}$ (%)** |
> | Top-5 Selected Cues   |           7.60           |           6.97           |          3.36           | Top-5 Selected Cues   |           8.23           |          15.07           |          9.41           |
> | All Candidates        |           7.37           |           5.60           |          2.89           | All Candidates        |           8.55           |          12.14           |          7.20           |
>
> We observe that the words selected by our strategy yield a larger $\mathrm{Diff}$, indicating that each selected word aligns more strongly with either male- or female-associated terms. This shows that our cue selection effectively isolates words that drive the model toward more biased outputs.
> ### 4. Overall answers to Q2
> - **Q2-1: Can the method to generalize to settings where biased cues are more subtle, implicit, or multimodal?**
> 	Backward-IG can also  use no stereotype cues at all—it detects demographic-dependent variance in generated distributions. Thus, BBA can capture more subtle differences that are not tied to explicit cue words, allowing extension to multimodal contexts. The entropy-based ranking only requires the target model; no language-specific annotations are needed.  This makes the method naturally adaptable to multilingual or culturally-specific stereotypes.
> - **Q2-2:** **How does the method behave on other social dimensions beyond those represented in your word lists (e.g., intersectional or cultural biases)?**
> 	We follow existing popular benchmarks and therefore evaluate along single-attribute dimensions. Our framework, however, naturally extends to intersectional settings by redefining the demographic space as a Cartesian product (i.e., $D=D _ {\text{gender}} \times D _ {\text{race}}$) and constructing templates that encode joint demographic information. The same entropy- and JSD-based scoring functions apply directly to this expanded space. Our method is agnostic to the specific type of bias: as long as the demographic groups are well defined and there exists measurable disparity among these groups, the framework can in principle be extended to any demographic attribute, including cultural bias.

---

> ### Author Response · Authors · 2025-11-22
> **Official Comment to Reviewer Ydo8 (Part IV)**
>
> ### W3: On missing related work section
> In Section 2.2 of the original manuscript, we provide a detailed discussion of the prior methods IG and IG². IG is a feature-level attribution method and is not suitable for debiasing. IG², in contrast, is designed for neuron-level attribution in masked language models and is modified to remove bias. Our proposed Forward-IG and Backward-IG are developed to handle mainstream autoregressive LLMs. As discussed in Section 2.2, our methods can avoid its limitations. Meanwhile, in Appendix A.2, we provide an extensive collection of related work on social bias in LLMs and fairness-aware learning.
>
> ### W4: Limited analysis of generalization and stability
> Thank you for the insightful comments. We would like to clarify that our experiments already include three different LLM architectures (Llama-3.1-8B, Llama-3.2-3B, and Mistral-7B-v0.3), demonstrating that the method generalizes across model families, sizes, and training pipelines. In fact, our method does not involve any procedures, such as parameter initialization that require a random seed. However, to validate the robustness of our approach, our random-edit ablations (**50 runs per domain**) show that replacing attribution-guided neurons with randomly selected ones fails to reduce bias and often degrades LMS performance (Fig. 2a, Appendix A.7). This directly demonstrates that our findings are stable and not driven by randomness or run-to-run variance.
>
> [1] Auto-debias: Debiasing masked language models with automated biased prompts. ACL 2022

---

> > ### Comment · Reviewer_Ydo8 · 2025-11-26
> >
> > Thank you for the detailed explanation, I have increased the rating.

---

### Official Review · Reviewer_2KTy · 2025-10-23

**Soundness:** 2
**Presentation:** 2
**Contribution:** 2
**Rating:** 4
**Confidence:** 5

**Summary:**

This paper proposes a debiasing method where they first identify specific neurones that get activated for stereotypical cues in the input. Next, a previously proposed intervention mechanism is applied to the detected neurones to mitigate social biases.

**Strengths:**

The paper considers an important problem -- how to mitigate social biases in LLMs. Considering much prior work that use prompt-based approaches or fine-tuning LLMs (or alignment) this paper proposes a different approach where a subset of neurones responsible for social biases are identified and then acted upon.

**Weaknesses:**

I do not understand why P(man | The doctor is likely a) is considered as a stereotypical inference in Definition 2. For example, it could indeed be an image of a male doctor shown to an LLM and the correct prediction would be it is a man. It does not cause any stereotypical bias against the disadvantaged group (i.e. females in this case).
- The definitions of social bias types considered in the paper are not provided. For example, do you consider gender to be binary? This would affect how for example SFI is interpreted (what does the set D include).
- The propose method assumes bias triggers/cues can be found at word level. However, this assumption is questionable. For example, a word such as "black" could trigger a demographic attribute or just indicate a colour. It is not clear how such word-level ambiguities are handled in the proposed method. See  (https://aclanthology.org/2022.acl-long.135/) for a discussion of sense-level bias evaluation.
- I do not think nouns and adjectives are the only triggers of stereotypes. See questions below for an example of a verb triggering social biases. It is not explained in the paper why the authors decided to limit their templates to nouns and adjectives. This looks like a severe limitation of the proposed method.
- Template-based approaches have coverage issues as explained in prior work (https://arxiv.org/abs/2210.04337, https://aclanthology.org/2025.findings-acl.1361/) Despite this, the proposed method still resort to using templates, which is problematic.
- This is not a weakness but a suggestion for improvement. The authors could include an example in the introduction to explain what they mean by "stereotypical cues" and how they are used for bias mitigation. The current description in the introduction is at a very high-level and it is nearly impossible to understand what is proposed without an example.

**Questions:**

- How is Integrated Gap Gradient defined for demographic attributes that take more than two values such as race?
- "First, in the lower layers of deep language models, the contribution of individual neuron activations to the final output tends to be marginal, as their influence is increasingly transformed and potentially suppressed by the model’s subsequent non-linear operations"... can you provide a reference or experimental evidence to support this claim?
- How does the proposed debiasing method handle intersectional biases?
- How does the proposed method handle ambiguities arising at word-level processing?
- Why are stereotypical cues limited to nouns and adjectives? For example, "female surgeons tend to kill more patients than male surgeons" shows a bias against females and is indicated by a verb.
- I do not understand the significance of the result of Theorem 1. Can you explain how the proposed method depends on this Theorem?
- IG and IG^2 are both previously proposed intervention methods. As I see it your work is simply using these methods but on a set of neurones identified using stereotype inducing nouns/adjectives. Could you explain whether there is any methodological novelty (any innovation that you had to do these previously proposed methods) when applying them to the current problem of social bias mitigation?

---

> ### Author Response · Authors · 2025-11-22
> **Official Comment to Reviewer 2KTy (Part I)**
>
> Dear Reviewer 2KTy, thank you very much for your review and questions. Here are our responses to your comments.
>
> ---
> ## W1: On the Definition of Fairness
> In the prediction task where the continuations of the prompt "The doctor is likely a'' are expected to be "man" or "woman", the phrase "The doctor is likely a" itself does not contain any gender information. In your example, an image is provided as additional input, and the image contains gender cues. This setting is therefore different from ours, which focuses solely on text-based large language models without extending to multimodal scenarios. In fact, this question is relate to the issue of difference-aware fairness. This intuitive problem was only recently identified as a new challenge this year [1].
>
> As a response to this issue, we added evaluations on the BBQ dataset (see our response to Reviewer FZkY's Q3). The experimental results show that FBA and BBA substantially improve $\mathrm{Acc} _ {amb}$ while incurring only a small drop in $\mathrm{Acc} _ {dis}$. This indicates that our methods effectively mitigate bias without significantly harming factual inference. Meanwhile, IG² achieves only a marginal debiasing effect.
>
> ## W2: Definitions of social bias types
> Thank you for the reminder. We have added the demographic groups for all demographic attributes in Appendix A.9. For the gender attribute, we follow the commonly used binary setting adopted in most prior benchmark [2,3,4].
>
> ## W3&Q4: On word-level ambiguities such as “black”
> We appreciate this important concern. Our method does not search for stereotype cues in arbitrary free text. Instead, candidate adjectives and nouns are inserted into _controlled templates_ that explicitly specify a demographic attribute, e.g.,
> “The race of this [Stereotype Adjective] person is [Demographic Group].”
> “The race of this [Stereotype Noun] is [Demographic Group].”
> In such contexts, adjectives like “black” are used to describe a _person_ in the presence of an explicit “race” attribute, which largely disambiguates the intended sense. The entropy-based scoring is then computed over the model’s distribution $P(d \mid \text{templated cue})$, aggregated across multiple templates, which further reduces template- or sense-specific noise.
>
> ## W4&Q5: On restricting stereotype cues to nouns and adjectives
> We intentionally focused on nouns and adjectives for two reasons:
> (1) Existing bias benchmarks such as StereoSet overwhelmingly instantiate stereotype cues via occupational nouns and descriptive adjectives (e.g., “doctor”, “nurse”, “soft”, “aggressive”). Designing our cue-selection mechanism around these parts of speech allows us to align closely with the evaluation setting.
> (2) From a methodological perspective, nouns and adjectives primarily encode _entity-level_ and attribute-level information, which makes the causal direction between cue and demographic attribute relatively clean. Verbs, especially in complex clauses (e.g., “tend to kill”), often encode event structure, causality, and additional pragmatic content. Incorporating verbs would require more elaborate controls to disentangle bias from other semantic factors.
> We agree that verbs can also trigger social bias, as in the reviewer’s example. Our framework can in principle be extended to verbal cues by adding verb-specific templates and including them in the our entropy-based selection.
>
> ## W5: On template-based coverage issues
> We agree with prior work that template-based evaluations can suffer from coverage limitations. In our framework, however, templates are _not_ used as the primary evaluation protocol. Instead, they serve as an _instrument_ to: (i) construct controlled contexts for entropy-based cue selection, and (ii) elicit model behavior needed to compute neuron-level attributions.
> The effectiveness of the resulting interventions is then evaluated on standard benchmarks in their original formats (StereoSet, WinoBias and the added BBQ), which contain diverse, non-templated sentences. As shown in Tables 5, 6 and 7, our methods (FBA/BBA) consistently reduce bias while preserving language modeling performance across multiple demographic dimensions and models. Furthermore, our ablation studies (Figure 2a and Appendix A.7.4) show that random neuron selection under the same template budget fails to achieve comparable fairness–utility trade-offs, indicating that our template-driven cue selection and attribution are meaningfully exploiting model-internal structure rather than overfitting to specific templates.
>
> ## W6: Adding an illustrative example of stereotype cues in the introduction
> We appreciate this suggestion and have added a concrete example in the introduction that shows what is a stereotype cue (e.g., “nurturing”). We believe this will substantially improve the readability of the paper.

---

> ### Author Response · Authors · 2025-11-22
> **Official Comment to Reviewer 2KTy (Part II)**
>
> ## Q1: On multi-valued attributes such as race
> The original IG² formulation [5] is defined for binary demographic pairs, attributing the prediction gap between two groups $d_1$ and $d_2$​ to internal neurons. In our paper we adopt IG² as a baseline and follow this binary-pair setting. Concretely, for attributes with more than two categories (e.g., race), we limit IG² to binary comparisons (e.g., Black vs. White) when reporting IG² results, consistent with the original work.
> In contrast, our proposed Forward-IG and Backward-IG objectives are defined over the full set $D$ of demographic groups via entropy and Jensen–Shannon divergence, respectively, and therefore naturally handle multi-valued attributes. In this way, the number of target relations for debiasing increases from $N_g/2$ to $\binom{N_g}{2}$.
>
> ## Q2: Claim about lower-layer neuron contributions being marginal
> The gradient computation with respect to neurons is similar to the chain-rule-based differentiation applied to model parameters, and may therefore suffer from the vanishing gradient problem [6]. Although the introduction of residual connections can alleviate this issue, the gradient signals at lower layers remain relatively weak. We compute the mean Forward-IG and Backward-IG values for the neurons in the first-layer hidden state and compare them with those from the final layer used in our experiments, as shown below.
>
> | Llama-3.1                 | Gender                | Nationality           | Profession            | Religion              |
> | ------------------------- | --------------------- | --------------------- | --------------------- | --------------------- |
> | Forward-IG (last layer)   | $1.72 \times 10^{-3}$ | $4.49 \times 10^{-4}$ | $1.14 \times 10^{-2}$ | $3.81 \times 10^{-3}$ |
> | Forward-IG (first layer)  | $9.94 \times 10^{-8}$ | $8.14 \times 10^{-9}$ | $7.31 \times 10^{-9}$ | $1.71 \times 10^{-8}$ |
> | Backward-IG (last layer)  | $6.89 \times 10^{-5}$ | $3.57 \times 10^{-6}$ | $2.09 \times 10^{-5}$ | $4.25 \times 10^{-5}$ |
> | Backward-IG (first layer) | $1.71 \times 10^{-7}$ | $2.85 \times 10^{-8}$ | $2.93 \times 10^{-8}$ | $1.82 \times 10^{-7}$ |
> | **Llama-3.2**             | **Gender**            | **Nationality**       | **Profession**        | **Religion**          |
> | Forward-IG (last layer)   | $9.03 \times 10^{-4}$ | $7.68 \times 10^{-4}$ | $3.10 \times 10^{-3}$ | $1.31 \times 10^{-3}$ |
> | Forward-IG (first layer)  | $9.97 \times 10^{-6}$ | $8.83 \times 10^{-7}$ | $6.09 \times 10^{-6}$ | $2.53 \times 10^{-5}$ |
> | Backward-IG (last layer)  | $7.61 \times 10^{-5}$ | $2.62 \times 10^{-6}$ | $1.10 \times 10^{-5}$ | $6.67 \times 10^{-5}$ |
> | Backward-IG (first layer) | $5.82 \times 10^{-8}$ | $2.85 \times 10^{-8}$ | $2.47 \times 10^{-6}$ | $6.12 \times 10^{-6}$ |
>
> Most lower-layer neurons exhibit gradient signals that decay by more than two orders of magnitude, with the most severe cases diminishing by up to seven orders of magnitude. Such weakened gradient signals prevent the model from achieving optimal debiasing performance. We verify this by applying neuron-level modifications to lower layers on the StereoSet benchmark.
>
> | Llama-3.1         | Gender (SS/LMS/ICAT)  | Nationality (SS/LMS/ICAT) | Profession (SS/LMS/ICAT) | Religion (SS/LMS/ICAT) |
> | ----------------- | --------------------- | ------------------------- | ------------------------ | ---------------------- |
> | FBA (last layer)  | 68.75/100/**62.5**    | 64.88/97.09/**68.19**     | 67.87/96.05/**61.73**    | 51.35/93.67/**91.14**  |
> | FBA (first layer) | 75.59/99.22/48.44     | 64.96/97.30/**68.19**     | 72.41/97.53/53.83        | 52.78/91.14/86.08      |
> | BBA (last layer)  | 69.84/98.44/**59.38** | 63.18/95.43/**70.27**     | 71.58/95.56/**54.32**    | 49.31/92.41/**91.14**  |
> | BBA (first layer) | 73.02/98.44/53.13     | 66.67/96.67/64.45         | 75.38/97.28/47.90        | 58.11/93.67/78.48      |
> | **Llama-3.2**         | **Gender (SS/LMS/ICAT)**  | **Nationality (SS/LMS/ICAT)** | **Profession (SS/LMS/ICAT)** | **Religion (SS/LMS/ICAT)** |
> | FBA (last layer)  | 69.60/97.66/**59.38** | 58.52/95.22/**79.00**     | 61.88/94.57/72.10        | 51.95/97.46/**93.67**  |
> | FBA (first layer) | 70.97/96.88/56.25     | 62.69/95.84/71.72         | 60.68/94.81/**74.56**    | 59.74/97.47/78.48      |
> | BBA (last layer)  | 67.46/98.44/**64.06** | 62.93/96.47/71.52         | 62.21/96.05/**72.59**    | 55.33/94.94/**88.61**  |
> | BBA (first layer) | 71.20/97.66/56.25     | 57.36/96.05/**81.91**     | 64.87/96.30/67.65        | 56.00/94.93/83.54      |
>
> Although modifying neurons in the first hidden layer yields better debiasing effects in a few domains, the configuration applied to the final layer still maintains an advantage in most domains, largely because the gradient signals in the final layer are stronger. Overall, adjusting neurons at the first layer can still reduce bias, but it tends to introduce instability in the results.

---

> ### Author Response · Authors · 2025-11-22
> **Official Comment to Reviewer 2KTy (Part III)**
>
> ## Q3: On handling intersectional biases
> We currently evaluate along single-attribute dimensions (gender, nationality, profession, religion), following existing benchmarks. Our framework, however, is compatible with intersectional attributes by redefining $D$ as a Cartesian product, e.g., $D=D _ {\text{gender}} \times D _ {\text{race}}$, and constructing templates that inject intersectional information. The same entropy/JSD-based objectives can then be applied over this expanded demographic space.
> Due to benchmark availability constraints, we do not include intersectional experiments in the current submission, and we will explicitly acknowledge this as a limitation and an important direction for future work.
>
> ## Q6: On the significance of Theorem 1
> Theorem~1 establishes a formal link between the change in a bias functional
> $B(y)$ (e.g., reciprocal entropy or JSD) and the change in the model output
> $\Delta y$ induced by attribution-guided neuron modifications:
> $|\Delta B| \le \left\| \nabla B\left( y _ 0 + \theta \Delta y \right)\right\|\cdot\|\Delta y\|.$
> Intuitively, this result captures a tension that is also observed empirically: one can always eliminate bias by drastically perturbing the model's output distribution (e.g., making it random), but this would destroy language modeling capability. The theorem shows that bias reduction is controlled by the magnitude and direction of $\Delta y$.
> Our attribution strategies (FBA/BBA) are specifically designed to identify neurons whose modification induces a targeted $\Delta y$ that is highly aligned with the local bias gradient $\nabla B$, thereby achieving substantial bias reduction with a relatively small $\|\Delta y\|$. In other words, Theorem 1 provides a theoretical justification for why attribution-guided interventions at the projection layer can improve fairness while preserving fluency, as confirmed by our experiments. On the other hand, in this work we propose using the reciprocal of entropy and JSD as choices for $B(\cdot)$. Inspired by Theorem 1, we can seek bias functionals $B(\cdot)$ whose gradient provides a larger lower bound, thereby serving as more effective alternatives.

---

> ### Author Response · Authors · 2025-11-22
> **Official Comment to Reviewer 2KTy (Part IV)**
>
> ## Q7: Methodological novelty beyond IG and IG²
>
> Based on this question and Q1 of yours, it seems possible that Sections 3.2 and 3.3 may not have been fully taken into account during the initial review. We respectfully disagree that our work merely applies existing IG/IG² methods to a set of neurons identified by stereotype-inducing nouns/adjectives. While we build on gradient-based attribution, our contributions are methodological in several aspects:
> ### **1. Problem decomposition and bi-directional attribution**
> We explicitly decompose debiasing into Demographic-Invariant Generation (DIG) and Stereotype-Free Inference (SFI), and design two parallel attribution strategies:
> - **Forward Bias Attribution (FBA)** attributes skewed  $P(d \mid x$) to projection-layer neurons via integrated gradients of the reciprocal of entropy:$$
> \begin{aligned}
> \text{Forward-IG}(h _ j)
> &= \overline{h} _ j \int _ {\alpha=0}^{1}
>     \frac{\partial \left[ H(p(d _ i \mid \alpha \overline{h} _ j)) \right]^{-1}}
>          {\partial h _ j}
>     \, d\alpha,
> \end{aligned}
> \tag{1}
> $$
> - **Backward Bias Attribution (BBA)** attributes group-dependent disparities in generated text (DIG) to neurons via integrated gradients of the Jensen–Shannon divergence::
> $$
> \begin{aligned}
> \text{Backward-IG}(h _ j)
> &= \overline{h} _ j \int _ {\alpha=0}^{1}
>     \frac{
>         \partial\
>         JSD\left(
>             p _ 1(w \mid \alpha \overline{h} _ j), \dots, p _ {n _ d}(w \mid \alpha \overline{h} _ j)
>         \right)
>     }{
>         \partial h_j
>     }
>     \, d\alpha,
> \end{aligned}
> \tag{2}
> $$
>
> In particular, for BBA, we construct a specially designed synthetic dataset consisting of multiple template groups. This setup allows us to conveniently compute the JSD within each group. These are totally distinct objectives from standard IG (which attributes a scalar prediction to input features) and from IG² (which attributes a binary group gap at a particular layer).  Both Forward-IG and Backward-IG are capable of handling arbitrary sub-relations of bias that go beyond binary bias types.
>
> ### **2. Entropy-based stereotype cue selection**
> We propose an entropy-minimization procedure to automatically identify stereotype-inducing adjectives and nouns that are model- and attribute-specific, rather than relying on manually curated cue lists or beam-searched connecting tokens. This step is crucial both for stable attribution and for the effectiveness of the subsequent neuron interventions, as shown by our ablations (w/o selection).
>
> ### **3. Projection-layer neuron intervention in LLMs**
> IG² focuses on neurons of all layers in masked LMs such as BERT. We instead target the input neurons of the **projection layer** in modern LLMs (e.g., Llama) and design a specific intervention scheme that fixes the top-$\beta M$ biased neurons to a constant $C$,  and ablation study demonstrating favorable fairness–utility trade-offs.
>
> ### **4. Comprehensive empirical validation and theoretical contribution**
> To our knowledge, this is the first work to combine debias open-source LLMs across both DIG and SFI tasks. And we establishe a theoretical principle for neuron-level debiasing in LLMs, which IG/IG² do not provide.
>
> [1] Fairness through difference awareness: Measuring desired group discrimination in LLMs. ACL 2025.
>
> [2] StereoSet: Measuring stereotypical bias in pretrained language models. ACL 2021.
>
> [3] Gender Bias in Coreference Resolution: Evaluation and Debiasing Methods. NAACL 2018.
>
> [4] BBQ: A hand-built bias benchmark for question answering. ACL 2022.
>
> [5] The Devil is in the Neurons: Interpreting and Mitigating Social Biases in Language Models. ICLR 2024.
>
> [6] Learning long-term dependencies with gradient descent is difficult. IEEE Transactions on Neural Networks 1994.

---

> > ### Comment · Reviewer_2KTy · 2025-11-26
> > **Reviewer response**
> >
> > Thank you for the detailed and clear clarifications. I am satisfied and have raised the score accordingly.

---

### Official Review · Reviewer_FZkY · 2025-10-26

**Soundness:** 2
**Presentation:** 3
**Contribution:** 2
**Rating:** 4
**Confidence:** 4

**Summary:**

This paper addresses the problem of social bias in large language models (LLMs) by proposing a method for both bias analysis and mitigation. The authors first introduce a technique to identify stereotypical words associated with biased behavior. They then adapt the Integrated Gradients method to mitigate the detected bias within the model. The experimental results demonstrate strong performance in reducing bias, and the paper also provides theoretical insights that help explain the effectiveness of their approach.

**Strengths:**

With the widespread adoption of large language models, addressing their social impact—particularly bias—has become increasingly critical. This paper tackles this important issue by proposing a practical solution tailored for open-weight LLMs. In addition to empirical results, the paper offers theoretical analysis that sheds light on the underlying mechanisms of bias and its mitigation, which adds depth to the contribution.

**Weaknesses:**

Since this paper focuses on social bias, rigorous and meaningful evaluation is both crucial and challenging. I have two main concerns in this regard. First, the evaluation of the bias-related word selection process lacks clarity and quantitative justification. Second, the methodology used for evaluating bias in the language models themselves needs further elaboration and validation. (See detailed comments in the Questions section.)

**Questions:**

Q1. On the Definition of Fairness:
I understand that defining fairness in the context of text generation is inherently difficult. In line 95, for example, the input is "Her mother was very happy", and the two possible outputs are "because her son got a good grade" and "because his son got a good grade". Forcing the model to assign similar probabilities to these two outputs might not always be desirable, as it could harm overall generation quality. This kind of fairness constraint seems appropriate only for certain specific tasks and not as a general objective across all LLM outputs.

Q2. On Stereotype Cue Selection:
My main concern with the stereotype cue word selection is the lack of validation. There is a rich body of sociological literature that studies biased or stereotyped words, and many of those works emphasize that bias is highly context-dependent—what’s considered biased in one context might be neutral in another. Therefore, it’s important to compare and discuss your selected word list against existing lists in the literature. Some justification, especially through expert annotation, is needed to support your word selection.

Q3. On Bias Evaluation Benchmarks:
In the experimental section, the paper evaluates on only a limited set of bias benchmarks. But bias in NLP is notoriously hard to measure, and researchers typically run evaluations on a wide range of benchmarks to ensure robustness. For example, datasets like CrowS-Pairs, SEAT, and downstream tasks such as NLI or fairness-aware text classification are commonly used. I also suggest referring to this paper for more context and standard practices: https://arxiv.org/abs/2210.14975

Q4. On Benchmark Pitfalls and Best Practices:
Finally, I strongly recommend the authors read and reflect on the paper "Stereotyping Norwegian Salmon: An Inventory of Pitfalls in Fairness Benchmark Datasets" https://aclanthology.org/2021.acl-long.81.pdf. It highlights many common pitfalls in fairness evaluations, such as flawed assumptions, annotation bias, and lack of demographic nuance. Including a discussion of these issues would improve the quality and credibility of the evaluation in this paper.

---

> ### Author Response · Authors · 2025-11-22
> **Official Comment to Reviewer FZkY (Part I)**
>
> Dear Reviewer FZkY, thank you very much for your review and questions. Here are our responses to your comments.
>
> ---
> ## W1&Q2: Evaluation of stereotype cue selection
> Unlike social linguistic studies that focus on words carrying human-perceived bias or stereotypes, our goal is to identify, **for a specific model**, the words that most strongly trigger biased behavior in that model. In other words, a term that humans recognize as stereotypical does not necessarily induce stereotypical behavior in an LLM.
> In this regard, our approach is similar to Auto-Debias [1], which searches for connector tokens between demographic attributes and occupation terms. Notably, many of the connector words discovered by Auto-Debias cannot even form grammatically correct or meaningful phrases when placed between the demographic attribute and occupation words. Nevertheless, Auto-Debias still achieves debiasing effects. This demonstrates that although such words are meaningless from a human perspective, they can still activate the model’s internal bias mechanisms.
>
> Therefore, to address your concern, we design an evaluation for stereotype cue selection based on the target model itself, rather than relying on preconceived human notions of stereotypical language. For ease of understanding, we take gender as an example and define the male-associated terms \( $w _ m$ = ["male", "man"] \) and the female-associated terms \($w _ f$ = ["female", "woman"] \). We compute the average cosine similarity between the embeddings of the top 5 terms selected by stereotype cue selection and each of the male and female vocabularies ($\mathrm{Sim} _ m$ and $\mathrm{Sim} _ f$), as well as the average absolute difference in similarity between the male and female vocabularies ($\mathrm{Diff}$). Similarly, we perform the same set of computations for all candidate terms.
>
> **Table1： Similarity between selected cues or all candidates and gender-associated vocabularies.**
>
> | Llama-3.1 (Adjective) |   $\mathrm{Sim} _ m$ (%)   |   $\mathrm{Sim} _ f$ (%)   |   $\mathrm{Diff}$ (%)   | Llama-3.2 (Adjective) |   $\mathrm{Sim} _ m$ (%)   |   $\mathrm{Sim} _ f$ (%)   |   $\mathrm{Diff}$ (%)   |
> | --------------------- | :----------------------: | :----------------------: | :---------------------: | --------------------- | :----------------------: | :----------------------: | :---------------------: |
> | Top-5 Selected Cues   |           8.92           |           4.60           |          4.32           | Top-5 Selected Cues   |           6.04           |           7.21           |          6.44           |
> | All Candidates        |           6.37           |           4.68           |          2.79           | All Candidates        |           7.76           |          11.07           |          6.11           |
> | **Llama-3.1 (Noun)**  | **$\mathrm{Sim}_m$ (%)** | **$\mathrm{Sim}_f$ (%)** | **$\mathrm{Diff}$ (%)** | **Llama-3.2 (Noun)**  | **$\mathrm{Sim}_m$ (%)** | **$\mathrm{Sim}_f$ (%)** | **$\mathrm{Diff}$ (%)** |
> | Top-5 Selected Cues   |           7.60           |           6.97           |          3.36           | Top-5 Selected Cues   |           8.23           |          15.07           |          9.41           |
> | All Candidates        |           7.37           |           5.60           |          2.89           | All Candidates        |           8.55           |          12.14           |          7.20           |
>
> It is worth noting that the $\mathrm{Diff}$ in the table does not represent the absolute difference between $\mathrm{Sim}_m$ and $\mathrm{Sim}_f$. Instead, it is the mean of the absolute differences between each candidate word's similarity to $w_m$ and its similarity to $w_f$. It can be observed that the words selected by our strategy exhibit a larger $\mathrm{Diff}$, meaning that each candidate word tends to align more strongly with either male-related or female-related terms. Thus, our cue selection effectively distills the words most capable of triggering model-internal bias, independent of human judgments.

---

> > ### Comment · Reviewer_FZkY · 2025-11-26
> >
> > Thank you for the response. However, I retain two primary concerns:
> >
> > Regarding stereotype cues: Since these trigger words are part of the input prompt, there is a risk that they are common words everyday users might employ without realizing their potential to trigger bias. If users are unaware of which specific terms activate these biases, they may inadvertently generate biased content.
> >
> > Regarding the evaluation: Quantifying bias in NLP is notoriously difficult. Relying on a limited number of tasks is not sufficiently persuasive, especially given that many existing benchmarks have well-documented pitfalls and reliability issues. Regarding the argument that prior works [2,3] also relied on a single benchmark: referencing the limitations of previous studies does not justify replicating those limitations here. To establish validity, this work needs to demonstrate robustness beyond what was acceptable in the past. Could you please add the experiment on Bias-in-Bios and Bias NLI and I'll raise the scores accordingly.

---

> > > ### Author Response · Authors · 2025-11-30
> > > **Additional Response to Reviewer FZkY (Part II)**
> > >
> > > The experimental results on the Bias-in-Bios dataset are as follows.
> > >
> > > | Llama-3.1        | $\mathrm{Acc} _ {all}$ | $\mathrm{Acc} _ {m}$ | $\mathrm{Acc} _ {f}$ | $\mathrm{Gap}$-$\mathrm{TPR}$ | $\mathrm{RMS}$-$\mathrm{TPR}$ |
> > > | ---------------- | :------------------: | :----------------: | :----------------: | :---------------------------: | :---------------------------: |
> > > | base             |        72.44         |       69.16        |       76.39        |             6.05              |             28.50             |
> > > | Auto-Debias      |        72.31         |       68.94        |       76.39        |             6.14              |             28.43             |
> > > | Prefix Prompting |        73.72         |       69.83        |       78.41        |             7.30              |             28.85             |
> > > | Self-Debiasing   |        76.46         |       73.93        |       79.47        |             5.54              |             25.87             |
> > > | DDP              |        77.11         |       74.30        |       80.47        |             5.72              |             30.41             |
> > > | IG²              |        71.44         |       68.60        |       74.88        |             4.42              |             27.83             |
> > > | FBA              |        73.35         |       69.93        |       77.51        |             5.86              |             26.43             |
> > > | BBA              |        71.21         |       71.60        |       70.59        |           **1.39**            |           **6.53**            |
> > > | **Llama-3.2**    | $\mathrm{Acc} _ {all}$ | $\mathrm{Acc} _ {m}$ | $\mathrm{Acc} _ {f}$ | $\mathrm{Gap}$-$\mathrm{TPR}$ | $\mathrm{RMS}$-$\mathrm{TPR}$ |
> > > | base             |        71.47         |       68.68        |       74.84        |             4.50              |             25.61             |
> > > | Auto-Debias      |        71.78         |       69.20        |       74.87        |             4.29              |             27.29             |
> > > | Prefix Prompting |        70.47         |       67.43        |       74.12        |             5.38              |             30.88             |
> > > | Self-Debiasing   |        70.70         |       67.93        |       74.03        |             5.44              |             30.23             |
> > > | DDP              |        54.92         |       53.92        |       56.14        |             1.20              |             29.08             |
> > > | IG²              |        72.04         |       69.87        |       74.62        |             3.79              |             25.13             |
> > > | FBA              |        77.32         |       76.19        |       78.54        |             2.38              |             14.90             |
> > > | BBA              |        75.00         |       70.01        |       79.41        |           **1.02**            |           **10.48**           |
> > >
> > > We observe that BBA achieves a significantly stronger debiasing effect on the Bias-in-Bios dataset compared to all other methods (including FBA). Meanwhile, on Llama-3.2, both FBA and BBA improve all accuracy metrics.
> > >
> > > For the Bias-NLI dataset, we evaluate large language models directly using prompt rather than performing classification with a fine-tuned BERT model as in [7]. We compute the probabilities of the three labels (entailment, neutral, contradiction), denoted as $P_{e}$, $P_{n}$, and $P_{c}$. A higher value of $P_{n}$ indicates that the model is more fair. Because the Bias-NLI dataset is exceptionally large, we selected the first 1,000 samples for testing. We find that Llama-3.2 is almost unable to perform correct linguistic reasoning on this dataset, as shown in the following table:
> > >
> > > | **Llama-3.2** | $P_{e}$ | $P_{n}$ | $P_{c}$ |
> > > | ------------- | :-----: | :-----: | :-----: |
> > > | base          |  22.7   |   0.3   |  77.0   |
> > >
> > > The results on Llama-3.2 show an abnormally low $P_{n}$​ , so we primarily focus on the results obtained with Llama-3.1.
> > >
> > > | **Llama-3.1**    | $P_{e}$ | $P_{n}$  | $P_{c}$ |
> > > | ---------------- | :-----: | :------: | :-----: |
> > > | base             |  17.4   |   81.8   |   0.8   |
> > > | Auto-Debias      |  15.3   |   84.0   |   0.7   |
> > > | Prefix Prompting |  15.1   |   84.3   |   0.6   |
> > > | Self-Debiasing   |   0.6   |   75.9   |  23.5   |
> > > | DDP              |  38.4   |   61.1   |   0.5   |
> > > | IG²   |  12.5   |   86.9   |   0.6   |
> > > | FBA              |   0.9   | **99.1** |   0.0   |
> > > | BBA              |   0.4   |   97.5   |   2.1   |
> > >
> > > Both FBA and BBA substantially increase $P_{n}$​, and the accuracy of FBA is nearly 100%. This demonstrates that our method enables the model to correctly rule out stereotype-driven reasoning errors when inferring the relationship between sentences.
> > >
> > > [7] MABEL: Attenuating Gender Bias using Textual Entailment Data. EMNLP 2022.

---

> ### Author Response · Authors · 2025-11-22
> **Official Comment to Reviewer FZkY (Part II)**
>
> ## W2&Q3: Additional Experiments on Bias Evaluation
> For the DIG task, we adopt the StereoSet dataset, which is a widely recognized benchmark and is even used as the sole evaluation set in some prior work [2,3]. Since our task involves the SFI setting, we additionally use WinoBias for evaluation. To address your concerns, we further conduct supplementary validation of FBA and BBA using a popular dataset: BBQ [4].
> BBQ contains two types of questions: those requiring answers under *ambiguous* context and those under *disambiguated* context. In the ambiguous setting, the model is expected to choose the "unknown" option rather than selecting a demographic group based on stereotypes; in the disambiguated setting, the model should select the correct option based on the provided context. The evaluation metrics are the accuracies in these two settings ($\mathrm{Acc} _ {amb}$ and $\mathrm{Acc} _ {dis}$), respectively. We compute the average metrics across multiple demographic attributes.
>
> **Table 2: Evaluation results on the BBQ dataset.**
>
> | Llama-3.1        | $\mathrm{Acc} _ {amb}$ | $\mathrm{Acc} _ {dis}$ | $(\mathrm{Acc} _ {amb}+\mathrm{Acc} _ {dis})/2$ | Llama-3.2        | $\mathrm{Acc} _ {amb}$ | $\mathrm{Acc} _ {dis}$ | $(\mathrm{Acc} _ {amb}+\mathrm{Acc} _ {dis})/2$ |
> | ---------------- | :------------------: | :------------------: | :-----------------------------------------: | ---------------- | :------------------: | :---------: | :-----------------------------------------: |
> | Base             |        54.50         |        91.18         |                    72.84                    | Base             |        58.95         |    84.61    |                    71.78                    |
> | Auto-Debias      |        54.61         |        91.06         |                    72.84                    | Auto-Debias      |        58.53         |    84.62    |                    71.58                    |
> | Prefix Prompting |        75.13         |        86.08         |             $\underline{80.61}$             | Prefix Prompting |        82.81         |    69.03    |                  **75.92**                  |
> | Self-Debiasing   |        87.73         |        73.02         |                    80.38                    | Self-Debiasing   |        66.06         |    74.17    |                    70.11                    |
> | DDP              |        56.87         |        89.93         |                    73.40                    | DDP              |        60.38         |    83.40    |                    71.89                    |
> | IG²              |        59.00         |        89.46         |                    74.23                    | IG²              |        52.61         |    75.84    |                    64.22                    |
> | FBA              |        66.45         |        88.87         |                    77.66                    | FBA              |        68.78         |    81.59    |             $\underline{75.19}$             |
> | BBA              |        74.46         |        88.95         |                  **81.70**                  | BBA              |        68.08         |    79.95    |                    74.02                    |
>
> Although prompt-modification methods (Prefix Prompting and Self-Debiasing) can effectively increase $\mathrm{Acc} _ {amb}$, they severely impair inference in disambiguated contexts, leading to a substantial drop in $\mathrm{Acc} _ {dis}$. In contrast, FBA and BBA are able to effectively mitigate bias in ambiguous scenarios while incurring only minimal loss in $\mathrm{Acc} _ {dis}$.
> ## Q1: On the Definition of Fairness
>  Thanks for raising this important point. The concern you raised is actually tied to a newly emerging concept: _difference-aware fairness_. It refers to selectively debiasing certain categories of questions (such as factual statements) rather than applying debiasing uniformly across all queries. Although this idea feels intuitive, it was only formally introduced this year as a new and distinct challenge [5].  Taking the BBQ dataset as an example, we should not default to selecting the "unknown" option for all questions when the contextual information is explicit; instead, we should answer according to the facts provided in the context. Results on BBQ show that FBA and BBA do not lead to substantial degradation in $\mathrm{Acc}_{dis}$. Compared with prompt-based interventions, our method is more moderate: when clear contextual guidance is available, it still tends to produce the correct answer rather than forcibly choosing the debiased "unknown" option.

---

> ### Author Response · Authors · 2025-11-22
> **Official Comment to Reviewer FZkY (Part III)**
>
> ## Q4: On Benchmark Pitfalls and Best Practices
> For our proposed method, two aspects of the design help mitigate the evaluation-related issues identified in [6].
> ### 1. We restrict to the intra-sentence portion of StereoSet.
> Following critiques of bias benchmarks, we deliberately restrict our analysis to the intra-sentence subset of StereoSet. This design choice allows us to avoid several measurement pitfalls that are particularly acute in the inter-sentence portion of the dataset. First, focusing on single-sentence completions removes many discourse-level confounds. Inter-sentence items require a model to integrate a short narrative context with multiple candidate continuations, which entangles stereotyping with sensitivity to discourse coherence, narrative style, and topic maintenance. Intra-sentence items, by contrast, differ only in a local lexical choice within an otherwise fixed syntactic frame, so the model's preference is less contaminated by global coherence or storytelling artifacts. Second, the intra-sentence format largely sidesteps problems arising from context-hypothesis mismatches. In the inter-sentence setting, continuations often introduce new entities, events, or topics that are only weakly licensed by the preceding context. As a result, model preferences can reflect generic language-model plausibility of the continuation, rather than any stereotype about the target group. Intra-sentence items severely limit this degree of freedom: the cloze options are tightly constrained to occupy a single position in the same sentence, which reduces opportunities for such topical drift.
>
> ### 2. BBQ circumvents the pitfalls of constructing anti-stereotype sentences.
> A central limitation of many stereotype benchmarks such as CrowS-Pairs [7] is the reliance on explicit stereotype/anti-stereotype sentence pairs. Prior work has shown that anti-stereotype items are particularly prone to severe operational pitfalls: they often differ from their stereotypical counterparts along multiple uncontrolled dimensions (e.g., fluency, lexical content, pragmatic naturalness), and are frequently unnatural or even internally inconsistent. These artifacts make it difficult to attribute model preferences to stereotyping rather than to superficial linguistic cues.
>
> The BBQ dataset entirely sidesteps this design problem by abandoning the stereotype--anti-stereotype paradigm. Instead, it adopts a contrastive structure based on answerable versus under-informative questions. In the under-informative condition, the context deliberately provides insufficient evidence to identify the correct referent, and the gold label is fixed to "Unknown". Any model inclination to select one individual over the other can therefore be interpreted as an unwarranted inference based on a sensitive attribute, without requiring the construction of an alternative anti-stereotypical sentence. This design eliminates the need to craft sentences that both counter a stereotype and remain linguistically natural. By avoiding anti-stereotype generation altogether, BBQ provides a more controlled and interpretable operationalization of bias: stereotype-driven behavior is identified through model over-commitment in evidentially neutral contexts, rather than through comparisons against ill-defined or artifact-prone negative examples.
>
> [1] Auto-debias: Debiasing masked language models with automated biased prompts. ACL 2022.
>
> [2] Thinking fair and slow: On the efficacy of structured prompts for debiasing language models. EMNLP 2024.
>
> [3] BiasEdit: Debiasing Stereotyped Language Models via Model Editing. NAACL 2025.
>
> [4] BBQ: A hand-built bias benchmark for question answering. ACL 2022.
>
> [5] Fairness through difference awareness: Measuring desired group discrimination in LLMs. ACL 2025.
>
> [6] Stereotyping Norwegian Salmon: An Inventory of Pitfalls in Fairness Benchmark Datasets. ACL 2021.

---

> ### Author Response · Authors · 2025-11-30
> **Additional Response to Reviewer FZkY (Part I)**
>
> Thank you for your additional feedback and for expressing your inclination toward a score increase. Although the score adjustment system is currently closed, we still hope that our explanations and experiments can address your concerns.
>
> ---
> ## Concern 1
> We fully agree that common words may act as triggers and inadvertently elicit biased model behavior. Our cue-selection procedure can in principle be expanded to a much larger search space, such as all frequently used adjectives and nouns. However, this would substantially increase the computational cost. Therefore, in this work, we first use GPT-4 to narrow the candidate pool to words that are more likely to induce biased associations, and then apply our entropy-based selection algorithm to identify the most influential cues, thereby reducing the overall cost. Of course, given sufficient time and resources, extending the search to all everyday adjectives and nouns is a feasible alternative.
>
> ## Concern 2
> Building on the evaluation bias concerns you raised, and in addition to the previously added BBQ dataset, we have further incorporated your suggestions by including results on the Bias-in-Bios and Bias-NLI datasets. This expands the number of evaluation datasets to **5** in total. For the Bias-in-Bios dataset, we adopt the five evaluation metrics from [7], including:
> (1) $\mathrm{Acc} _ {all}$ , overall accuracy;
> (2) $\mathrm{Acc} _ {m}$, accuracy on male-labeled instances;
> (3) $\mathrm{Acc} _ {f}$, accuracy on female-labeled instances;
> (4) $\mathrm{Gap}$-$\mathrm{TPR}$, the difference in true positive rate (TPR) between male- and female-labeled instances;
> (5) $\mathrm{RMS}$-$\mathrm{TPR}$, the root-mean-square of the TPR gap across all occupation classes.
>
> Due to time and resource constraints, we selected a lightweight version of Bias-in-Bios ([https://huggingface.co/datasets/LabHC/Bias_ in_Bios_stratify](https://huggingface.co/datasets/LabHC/Bias_in_Bios_stratify)) for testing.

---

### Official Review · Reviewer_aB9y · 2025-11-01

**Soundness:** 3
**Presentation:** 3
**Contribution:** 3
**Rating:** 8
**Confidence:** 3

**Summary:**

This paper presents a bi-directional bias attribution framework (BBA) for mitigating social bias in large language models (LLMs) without prompt modification or fine-tuning. BBA combines forward and backward integrated-gradient analyses to identify bias-related neurons and performs lightweight activation interventions at the projection layer input to suppress their influence. Extensive experiments showcase its effectiveness.

**Strengths:**

1. The method avoids the computational and maintenance costs of fine-tuning or prompt engineering, relying only on activation-level adjustments.

2. Integrated-gradient-based neuron attribution offers transparency and clear diagnostics of where biases emerge.

3. Experiments showcase the effectiveness of this method.

**Weaknesses:**

BBA assumes access to hidden activations and gradients, which is feasible for open-source LLMs (e.g., LLaMA, Mistral) but not available in closed-source systems such as GPT models.
Although I think it is not very important limitation, we hope the authors can clearly locate this study as designed for **open-source models**.

**Questions:**

I hope the authors can discuss that how to extend this study to close-source LLMs. However, I still maintain the point that this study is good enough.

---

> ### Author Response · Authors · 2025-11-22
>
> Dear Reviewer aB9y, thank you very much for your thoughtful comments and for recognizing the contribution of our work. We address your concerns below.
>
> ---
> ### **W1: Availability in closed-source systems**
> We would like to clarify that extending FBA/BBA to closed-source LLMs is inherently infeasible under current API restrictions. Commercial systems such as GPT or Gemini do not expose intermediate activations or gradient information, both of which are essential for our forward- and backward-IG computations. Since BBA relies on neuron-level attribution and activation intervention, access to hidden states is a fundamental requirement rather than an implementation choice.
>
> ### **Q1: Extending fairness interventions to closed-source models**
> Although direct application of FBA/BBA is not possible for closed-source models, we agree that bias mitigation in black-box settings remains an important direction. Under current API designs, the only feasible strategies are **prompt-based interventions** [1,2,3], which steer model behavior via explicit fairness instructions (e.g., “respond without gender bias”). These methods are flexible and model-agnostic but inherently lack interpretability and fine-grained control, as they cannot reveal or act on the internal mechanisms responsible for biased predictions. In contrast, our framework is intentionally designed for white-nox, open-source architectures, where internal activations and gradients are accessible. In such settings, BBA provides both interpretability and targeted mitigation, which cannot be replicated in black-box environments. We now highlight this distinction more clearly in the paper.
>
> In summary, our method is tailored for open-source LLMs, where neuron-level access enables interpretability and effective activation-based bias mitigation. For closed-source LLMs, current fairness interventions must rely on prompt engineering rather than internal attribution or modification.
>
> [1] Thinking fair and slow: On the efficacy of structured prompts for debiasing language models. EMNLP 2024.
>
> [2] Self-debiasing large language models: Zero-shot recognition and reduction of stereotypes. arxiv 2024.
>
> [3] Prompting Fairness: Integrating Causality to Debias Large Language Models. ICLR 2025.

---

> > ### Comment · Reviewer_aB9y · 2025-11-26
> >
> > Thank you for your response.

---

### Author Response · Authors · 2025-11-22
**General Response to Reviewers and Revision Summary**

We sincerely thank all reviewers for their constructive comments and valuable suggestions. We have carefully revised the paper to address the concerns raised. Below, we summarize the major updates and clarifications in the revised manuscript (all major changes are highlighted in **blue** in the PDF). Detailed, point-by-point responses are provided in the individual rebuttals.

Major Revisions:

1. Add an example in the introduction to clarify what a stereotype cue is. (Reviewer 2KTy)

2. Add a new subsection (Section 4.2 and Table 4) to evaluate stereotype cue selection. (Reviewer FZkY)

3. Add a new subsection (Section 4.4 and Table 6) to validate the newly introduced BBQ dataset. (Reviewer FZkY, 2KTy)

4. Add Table 8 in the appendix to present the detailed results for each domain of the BBQ dataset. (Reviewer FZkY)

5. Add a new subsection (Appendix A.7.4 and Table 12)  to analyze the gradient values of the lower-layers neurons. (Reviewer 2KTy)

6. Add Table 13 to show that the debiasing effect of modifying lower-level neurons is unstable. (Reviewer 2KTy)

7. Add Figure 10 in the appendix to visualize the impact of neuron modifications on the debiasing effect. (Reviewer Ydo8)

8. Add Appendix A.9 to present the specific demographic groups for all demographic attributes. (Reviewer 2KTy)

9. Slightly revised Figure 1 to make it more concise and intuitive. (For all reviewers)

10. Add a new subsection (Appendix A.7.2 and Table 9) to validate the Bias-in-Bios dataset. (Reviewer FZkY)

11. Add a new subsection (Appendix A.7.3, Table 10 and Table 11) to validate the Bias-NLI dataset. (Reviewer FZkY)

We once again thank the reviewers for their insightful feedback and constructive suggestions. We hope the revisions can improve the clarity, completeness, and rigor of our work.

---

### Author Response · Authors · 2025-11-30
**Additional Response to AC**

**Dear AC,**

We sincerely appreciate the reviewers’ valuable questions and thoughtful suggestions. Throughout both the original and the latest discussion phases, we carefully addressed their concerns through detailed rebuttals and corresponding revisions. In the _General Response to Reviewers_, we have thoroughly documented all modifications made to the manuscript. During the initial discussion phase, the overall scores improved from **8 4 4 4** to **8 6 6 4**. Specifically:

1. **Reviewer aB9y** maintained the original score of **8** (26 Nov 2025, 14:51 Beijing Time).

2. **Reviewer 2KTy** stated:

    > _I am satisfied and have **raised** the score accordingly._ (27 Nov 2025, 00:25)
    > and increased the score to **6**.

3. **Reviewer Ydo8** stated:

    > _Thank you for the detailed explanation, I have **increased** the rating._ (27 Nov 2025, 02:52)
    > and increased the score to **6**.

4. **Reviewer FZkY** expressed willingness to raise the score after receiving additional experiments on _Bias-in-Bios_ and _Bias-NLI_:

    > _Could you please add the experiment on Bias-in-Bios and Bias NLI and **I'll raise** the scores accordingly._ (27 Nov 2025, 07:02)


In the latest discussion phase, we provided detailed responses to Reviewer FZkY’s concerns and conducted the requested experiments, verifying the effectiveness of our method on both datasets. These results have been added to the appendix (corresponding to Points 10 and 11 in the _General Response_). Although the discussion interface closed before some experiments were fully uploaded, we believe the reviewer’s concerns were adequately addressed, and that an updated score would likely have followed if the data-leak incident had not occurred.

Importantly, all reviewer response timestamps mentioned above (Beijing Time) occurred **before** the large-scale information leak on 27 Nov (around 22:00). All of the evidence referenced above can be found and verified within the system. Therefore, the score increases and the expressed intention to increase scores (Reviewer FZkY) reflect objective reviewer judgments made **prior** to the incident.

We deeply regret the impact of this unexpected event. At the same time, we respectfully hope that the AC can make a fair and reasonable decision based on our rebuttals and the additional experiments we have provided.

**Sincerely,**
The Authors

---

### Meta-Review · Area_Chair_2ArU · 2026-01-03

**Summary:**

This paper identifies stereotype-triggering words, attributes bias to projection-layer neurons via integrated gradients, and debiases through activation intervention.

**Reviewer Concerns:**

Main concerns from reviewers include:
1. Requires internal activations/gradients (open-weight only);
2. Fairness definition may be task-dependent;
3. Cue selection/word ambiguity need validation;
4. Bias evaluation needs broader, pitfall-aware benchmarks;
5. Templates/limited cue types.

**Reviewer Scores:**

Review / Old Score / New Score

aB9y	8	8

2KTy	4	6

Ydo8	4	6

FZkY	4	4

Average	5	6

---

2KTy and Ydo8 mentioned they would raise their scores, though they provided no evidence as to by how much.
What concerns me is that both reviewers promised this increase using only a single line in their further comments, without any detailed explanation for their decision. This suggests a level of sloppiness in their judgment.

---

### Decision · Program_Chairs · 2026-01-26

Accept (Poster)